# Four New Species of Zosimeidae (Copepoda: Harpacticoida) from the Southwestern Gulf of Mexico

Jisu Yeom [1,2], Melissa Rohal Lupher [3,4] and Wonchoel Lee [1,2,*]

1 Department of Life Science, College of Natural Sciences, Hanyang University, 222 Wangsimni-ro, Seoul 04763, Korea; ymjisoo92@gmail.com
2 Research Institute for Natural Sciences, Hanyang University, 222 Wangsimni-ro, Seoul 04763, Korea
3 Texas Water Development Board, 1700 North Congress Avenue, Austin, TX 78701, USA; melissa.lupher@twdb.texas.gov
4 Harte Research Institute for Gulf of Mexico Studies, Texas A&M University-Corpus Christi, 6300 Ocean Drive, Unit 5869, Corpus Christi, TX 78412, USA
* Correspondence: wlee@hanyang.ac.kr; Tel.: +82-2-2220-0951

**Abstract:** As a part of ongoing efforts for monitoring benthic ecosystem in the Gulf of Mexico, Harpacticoid copepods were collected from the southwestern Gulf of Mexico. Among them we report three new species of *Zosime*, and a new species of *Peresime* for the study area. *Zosime* is the most specious-rich genus in the family. Three species of *Zosime* were morphologically similar to *Z. paratypica* Becker and Schriever, 1979, *Z. atlantica* Bodin, 1968, and *Z. destituta* Kim J.G., Jung and Yoon, 2016, respectively. However, all three species have unique characteristics that distinguish them from similar species such as setal formulae and shape of female P5 and caudal rami. The new species of *Peresime* has similar morphological characters with *P. reducta* (Becker and Schriever, 1979). These two species can be distinguished by differences in mouth parts and the length of the setae on P2. This is the first report on the genera *Zosime* and *Peresime* from the Gulf of Mexico. We also discuss the global diversity and distribution of Zosimeidae and provide a key to the genera of the family and species of each genus, including the four new species from the Gulf of Mexico.

**Keywords:** taxonomy; meiofauna; diversity; deep sea; Tisbidae

## 1. Introduction

The Gulf of Mexico has undergone two major oil spills [1] and has been the subject of several environmental studies focused on planktonic and benthic organisms [2–6].

The number of harpacticoid copepods reported in the Gulf of Mexico is small, with only 71 species listed [4]. However, a total of 696 harpacticoid species were identified during a survey in the northern Gulf of Mexico [3] implying they have a high diversity in the deep sea. Species of Zosimeidae are dominant in the harpacticoid community of the northern Gulf of Mexico. Zosimeidae are responsible for 32.98% of contribution of harpacticoid families to total harpacticoid abundance [3]. Although Baguley et al. [3] reported that "Tisbidae" has the highest contribution in the northern Gulf of Mexico, most of the species in their study actually belong to Zosimeidae (personal observation by W.L.).

The family Zosimeidae is a small family harboring five genera, *Zosime* Boeck, 1873, *Peresime* Dinet, 1974, *Pseudozosime* Scott T., 1912, *Acritozosime* Kim, JG and J Lee, 2021, and *Heterozosime* Kim, JG and J Lee, 2021. Currently each genus consists of 23, 2, 1, 1, 1 valid species, respectively. Koller and George [7] summarized the distribution of 15 species of *Zosime*, and Kim et al. [8] provided a key to the species of the genus known in year 2016. In this study, we provide a key to the genera of the family and to the species of each genus, along with the descriptions of four new species of two genera. In addition, the contents related to molecular research and distribution of Zosimeidae, and the morphological features of species within *Zosime* were summarized (Table S1). This is the first report on the genera *Zosime* and *Peresime* from the Gulf of Mexico.

## 2. Materials and Methods

### 2.1. Sample Collection

Meiofaunal harpacticoids were collected from four stations in the southern Gulf of Mexico (Figure 1). The stations were part of a study designed to determine the recovery rate of the benthic community following an oil spill. Samples were collected onboard the Universidad Nacional Autonoma De Mexico's R/V Justo Sierra, from 30 July to 9 August, 2015. The samples were collected around the Ixtoc-1 wellhead, which was the site of the 1979 well blow out [9].

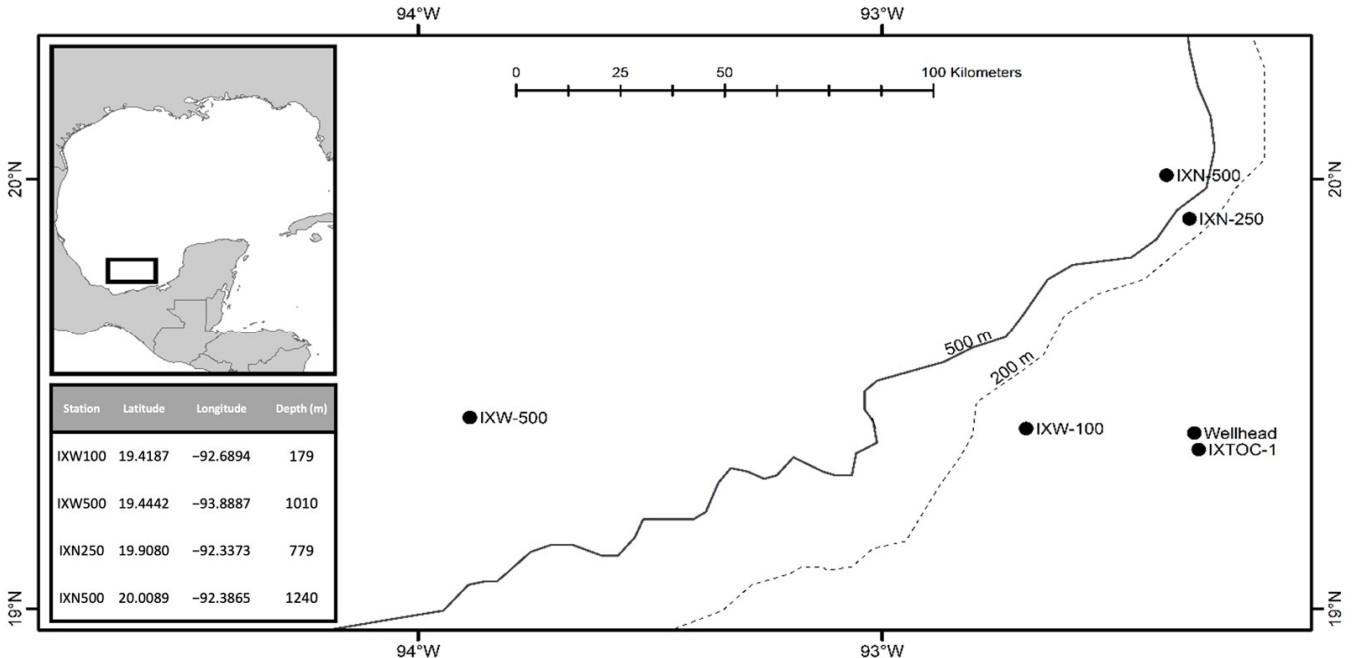

**Figure 1.** Map showing the location of the sample stations and depth.

Sediments were sampled with a multiple corer. Three replicate cores with an inner diameter of 9.5 cm were collected at each station. The sediment samples were fixed with 7% formalin buffered with Borax©. Copepods were extracted from the sediment samples by sieving the samples using 45 μm mesh sieve and then removed by hand and preserved in 70% ethanol.

### 2.2. Morphological Examination

Specimens were dissected in lactic acid and dissected parts were mounted on slides in lactophenol as a mounting medium. Preparations were sealed with transparent nail varnish. All drawings were prepared using a drawing tube attached to a Zeiss Axioskop phase-interference compound microscope and Olympus BX51 differential interference contrast microscope.

The descriptive terminology of Huys et al. [10] was adopted. Abbreviations used in the text are as follows: A1, antennule; A2, antenna; ae, aesthetasc; exp, exopod; enp, endopod; P1–P6, first to sixth thoracopod; exp (enp)-1 (2, 3) to denote the proximal (middle, distal) segment of a three-segmented ramus. Specimens were deposited in the National Marine Biodiversity Institute of Korea (MABIK). Scale bars in figures are in μm.

### 2.3. Phylogenetic Analysis

The molecular analysis involved 21 nucleotide sequences. All sequences were down-loaded from GenBank (Table S2). Obtained sequences were aligned by the ClustalW algorithm [11] in MEGA version 7.0 [12]. Phylogenetic analyses were performed using Maximum Likelihood (ML) approaches. ML analysis used the K2 (Kimura 2-parameter

model, [13]) +G+I model based on the model test result in MEGA. A discrete Gamma distribution was used to model evolutionary rate differences among sites (5 categories (+G, parameter = 0.5681)). The rate variation model allowed for some sites to be evolutionarily invariable ((+I), 38.47% sites). One thousand bootstrap replicates were performed to obtain a relative measure of node support for the resulting trees. Tree was rooted with *Tigriopus* (Harpacticidae) sequences.

## 3. Results

*3.1. Systematics*

Subclass Copepoda Milne Edwards, 1840
Order Harpacticoid G. O. Sars, 1903
Family Zosimeidae Seifried, 2003
Genus *Zosime* Boeck, 1873
*Type species: Zosime typica* Boeck, 1873.
*Zosime montagnai* sp. nov.
(Figures 2–5)

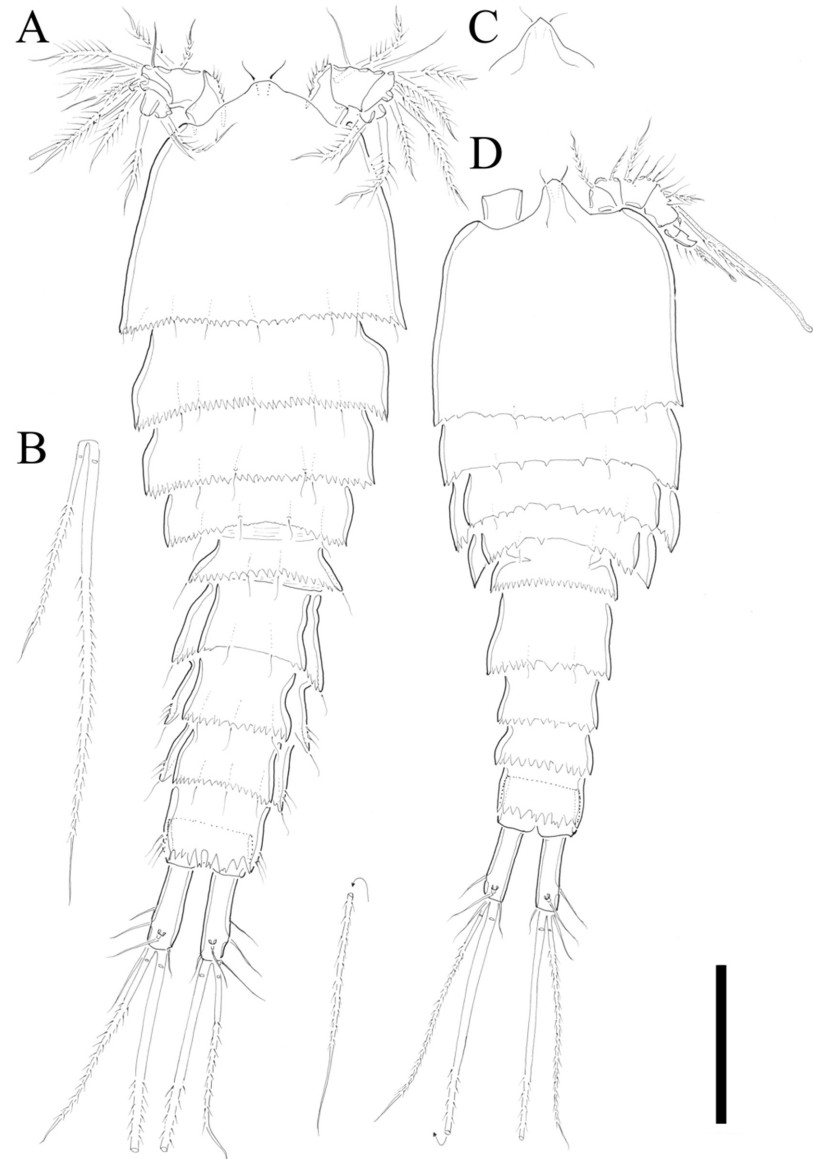

**Figure 2.** *Zosime montagnai* sp. nov. Female. (**A**) Habitus, dorsal; (**B**) Causal seta VI and V; (**C**) rostrum at different angle. Male. (**D**) Habitus, dorsal. Scale bar: 100 µm.

*Type Locality.* IXN250 station (19°54′28.8″ N, 92°20′14.3″ W) in the southern Gulf of Mexico, north-west Atlantic Ocean (depth: 779 m).

*Material Examined.* Holotype: 1♀(MABIK CR00249456), Paratypes: 1♂(MABIK CR00249457) from IXW500, 1♀(MABIK CR00249458) from IXN250.

*Etymology.* The new species is dedicated to Prof. Paul A. Montagna (Harte Research Institute for Gulf of Mexico Studies, Texas A&M University—Corpus Christi) for his excellent contributions to the study of harpacticoid copepods and the benthic community in the Gulf of Mexico. It is a noun in the genitive case, gender masculine.

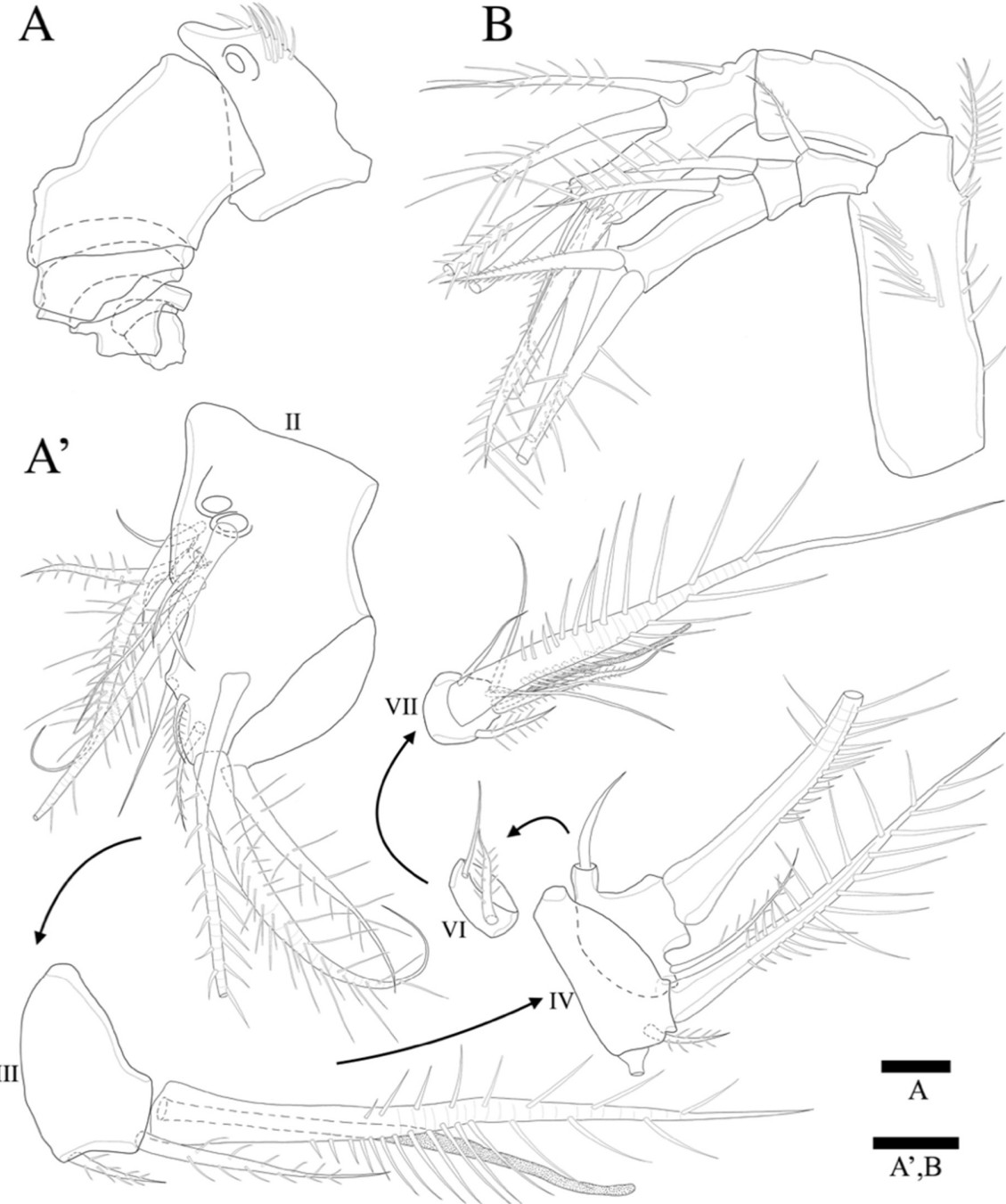

**Figure 3.** *Zosime montagnai* sp. nov. Female. (**A**) A1 showing segmentation, armature omitted (**A′**) A1; (**B**) A2. Scale bars: 10 μm.

*Differential Diagnosis*. Rostrum triangular, slightly pointed apically. Each somite armed with serrated posterior margin. Antennule seven-segmented in both sexes. Genital somite and succeeding urosomal segments with lateral expansions in the female. P3- and P4-bearing somites with lateral expansion on the male. Caudal rami three times longer than wide in both sexes. Female P5 with six setae, without seta on exp surface. Male P2 endopod two-segmented, and enp-2 modified into hook shaped apophysis with two pinnate setae.

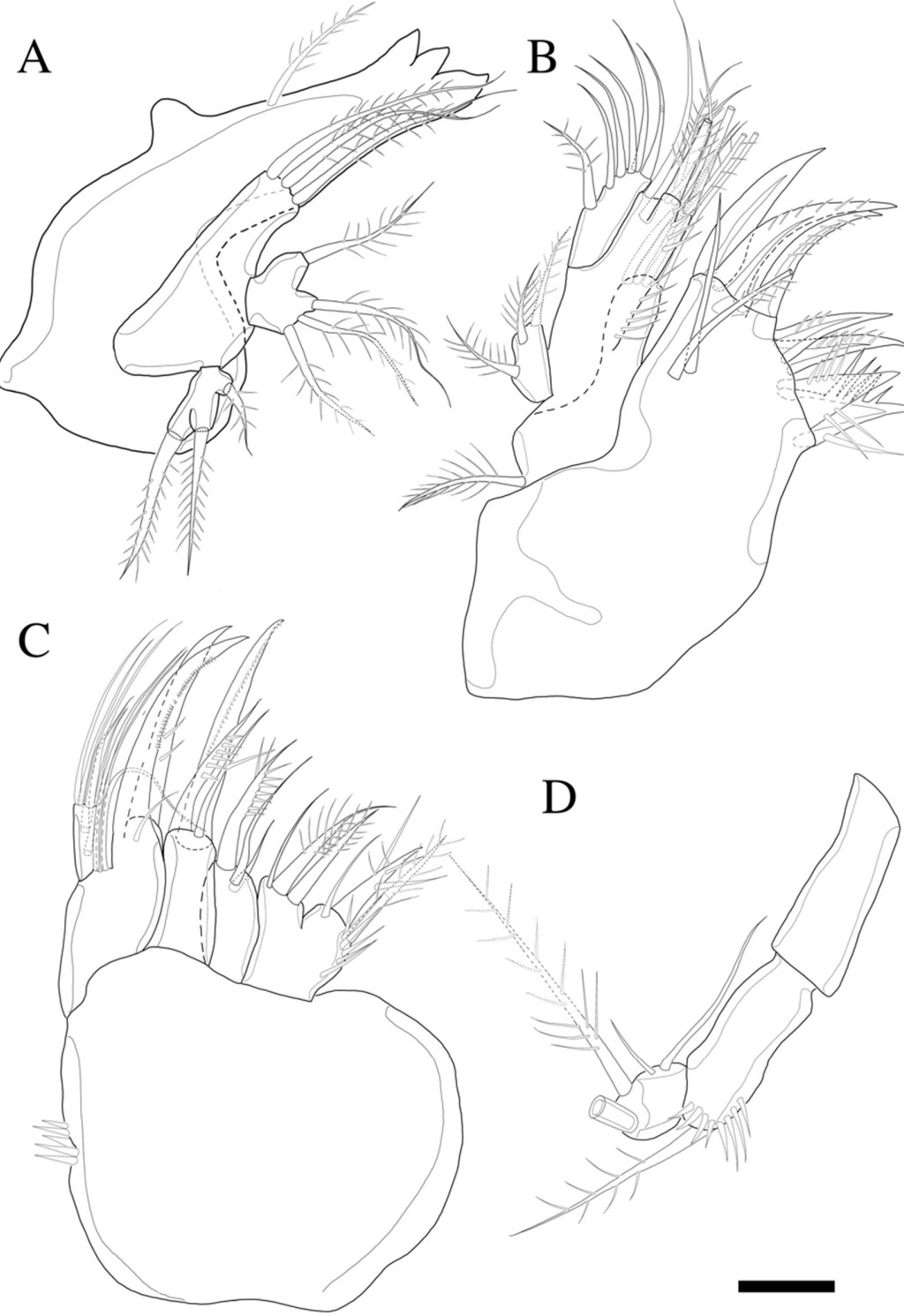

**Figure 4.** *Zosime montagnai* sp. nov. (**A**) Mandible; (**B**) Maxillule; (**C**) Maxilla; (**D**) Maxilliped. Scale bar: 10 μm.

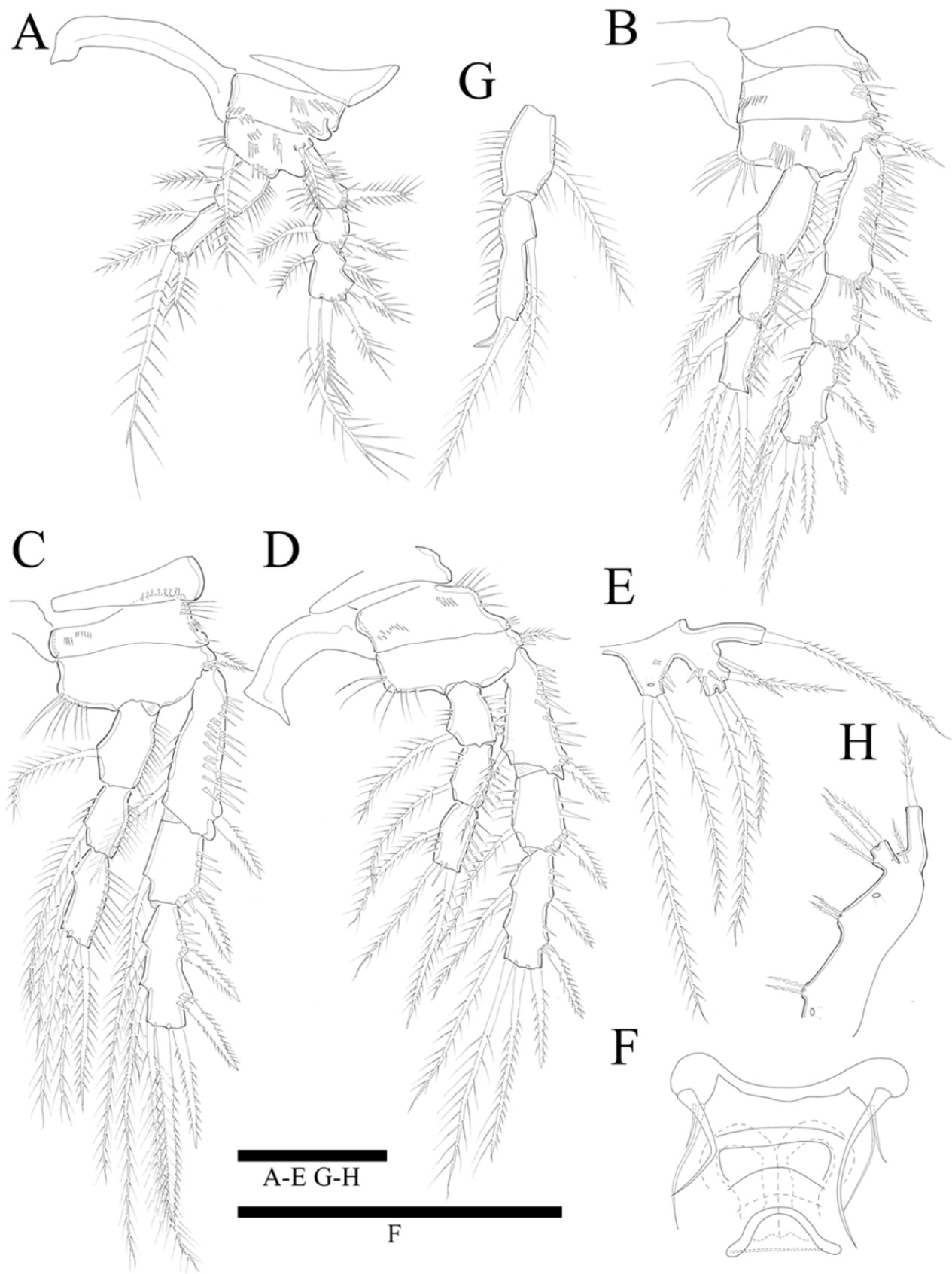

**Figure 5.** *Zosime montagnai* sp. nov. Female. (**A**) P1; (**B**) P2; (**C**) P3; (**D**) P4; (**E**) P5; (**F**) P6. Male. (**G**) P2 endopod; (**H**) P5. Scale bars: 50 μm.

*Description of Female.* Total body length of holotype 535 μm (measured from anterior margin of rostrum to posterior margin of caudal rami). Largest width (175 μm) measured at posterior margin of cephalic shield. Urosome distinctly narrower than prosome (Figure 2A).

Cephalothorax triangular with serrate posterior margin; dorsal surface smooth with few sensilla posteriorly. Prosomites (Figure 2A) with smooth dorsal surface, serrated posterior margins, and few sensilla posteriorly as figured.

Rostrum fused to cephalothorax, triangular, and slightly pointed apically, with two sensilla (Figure 2A,C).

Urosomites (Figure 2A) with serrated posterior margin as illustrated. Dorsal surface and posterior margin of P5-bearing somite ornamented as preceding somites. Genital double-somite with smooth dorsal surface. Genital somite and succeeding urosomal segments produced laterally. Each urosomite with several posterior sensilla as illustrated (Figure 2A).

Anal somite (Figure 2A) completely covered with well-developed pseudoperculum, unornamented.

Caudal rami (Figure 2A,B) twice longer than anal somite, and three times longer than wide; setae I and II arising laterally halfway along outer margin, seta II longer than seta I; seta III longer than setae II; seta IV basally fused to seta V; seta V longest, much longer than all urosomites combined; seta VI as long as seta I and located on distal inner corner; seta VII bare, located dorsally, triarticulated.

Antennule (Figure 3A,A') seven-segmented. Segment 1 with row of strong spinules around proximal margin. Segment 2 largest. Segment 3 with aesthetasc fused basally to strong pinnate seta. All setae pinnate except for three, one, one, and four naked setae on second, fourth, fifth, and last segments, respectively. Armature formula: 1-(1 pinnate), 2-(5+9 pinnate), 3-(2 pinnate+(1+ae)), 4-(1+2 pinnate), 5-(1+2pinnate), 6-(1+1pinnate), 7-(4+1 pinnate+1 acrothek). Apical acrothek consisting of one well-developed but small ae fused basally to strong and stout pinnate seta, and bare seta.

Antenna (Figure 3B) four-segmented, comprising coxa (not figured), basis, and free two-segmented endopod. Basis with several rows of spinules along inner margin, with pinnate abexopodal seta subdistally. First endopodal segment with a subdistal seta missed during dissection and sample preparation (the scar indicates the position of a seta); segment two with row of stout spinules on apical margin; apical armature consisting of two stout pinnate spines and naked spine, slender distal pinnate seta and two distal geniculate spines (damaged during dissection); and two inner lateral pinnate spines. Exopod three-segmented, with one, one, and three setae, respectively; all setae and spines pinnate; second segment shortest; first segment twice as long as second; last segment much longer than preceding two segments combined, with one lateral and two apical pinnate setae.

Mandible (Figure 4A) well-developed, gnathobasis with three teeth and pinnate seta at distal corner. Basis with three pinnate setae. Exopod one-segmented with one lateral and two apical pinnate setae. Endopod one-segmented, with one lateral and three apical pinnate setae.

Maxillule (Figure 4B) with praecoxa without spinular ornamentation along outer lateral margin. Arthrite strongly developed, with two surface setae (one naked, one pinnate), seven elements around distal margin. Coxa with cylindrical endite bearing four setae apically, and epipodite on outer lateral margin. Basis with three pinnate and three naked setae. Endopod one-segmented with one pinnate, and five naked setae. Exopod one-segmented, smaller than endopod with three pinnate setae (inner most one missing in Figure 4B).

Maxilla (Figure 4C). Syncoxa with spinular row on anterior surface and three endites; proximal endite bilobate, proximal lobe with pinnate spine, pinnate seta, and naked seta, and distal lobate with one naked and two pinnate setae; middle endite with pinnate spine and two naked setae; distal endite with stout serrate spine and two pinnate setae. Allobasis produced into strong curved claw, and pinnate curved spine with short slender seta basally; accessory armature consisting of three slender lateral setae proximally, and close to base of endopod. Endopod one-segmented with five slender distal setae.

Maxilliped (Figure 4D). Syncoxa elongate and cylindrical without ornamentations. Basis with row of outer spinules distally, with pinnate seta distally. Endopod small with two long, sparsely pinnate seta (one seta broken in Figure 4D); accessory armature consisting of two naked setae.

Swimming legs 1–4 (Figure 5A–D) biramous, with three-segmented exopods, and P1 with two-segmented endopod, P2–P4 with three-segmented endopods and with wide intercoxal sclerites, and well-developed triangular praecoxae. Coxae and bases with anterior rows of surface spinules as illustrated, the former rectangular.

P1 (Figure 5A). Basis with strong inner pinnate seta, and with spinules along inner margin, and with outer pinnate seta, and several spinules along outer margin. Exopod three-segmented; exp-1 and exp-2 with pinnate outer spine, respectively; exp-2 with inner pinnate seta; exp-3 with three pinnate outer spines and two pinnate distal setae and inner pinnate seta. Endopod two-segmented; enp-2 slightly longer than enp-1, and with two inner setae and two apical spines; inner apical spine four times longer than outer one.

P2 (Figure 5B). Basis with row of long spinules along inner distal margin, and with inner pinnate seta, and several spinules along outer margin. Exopod longer than endopod; exp-1 longest and exp-2 shortest; exp-1 and exp-3 with, exp-2 without row of inner spinules. Endopod three-segmented; enp-1 longest, enp-2 and enp-3 subequal in length; each segment with outer row of spinules; enp-3 reaching only distal 1/3 of exp-3.

P3 (Figure 5C). Basis with row of long spinules (slightly thinner than those in P2) along inner distal margin, and with outer pinnate relatively short spine, and few scattered spinules along outer margin. Exopod longer than endopod; exp-1 longest and exp-2 shortest; exp-1 with row of inner setules; each endopod segment with row of long outer spinules; enp-2 smallest; enp-1 and enp-3 subequal in length; endopod only reached to proximal 1/3 of exp-3.

P4 (Figure 5D). Coxa trapezoid, distal margin longer than proximal margin. Basis with row of long spinules along inner distal margin, and with outer pinnate seta, and several spinules along outer margin. Exopod longer than endopod; all exopod segments with row of inner spinules; exp-3 longest, and exp-2 shortest. Each endopod segments with row of spinules along outer and inner lateral margins; all segments subequal in length; enp-3 with well-developed an apical tube pore between two apical setae; endopod reached to proximal region of exp-3. Armature formulae as in Table 1.

**Table 1.** Armature formulae of legs 1–4.

|  | Exopod | Endopod |
| --- | --- | --- |
| P1 | 0.1.123 | 1.220 |
| P2 | 1.1.223 | 1.1.121 |
| P3 | 1.1.223 | 1.1.121 |
| P4 | 1.1.223 | 1.1.120 |

P5 (Figure 5E). Outer basal seta long and pinnate set on cylindrical setophore. Exopod fused to baseoendopod. Endopodal lobe trapezoid with pore near apical region, and two long pinnate apical setae of which the inner is longer than the outer; with few scattered anterior spinules along outer lateral margin. Exopod rectangular, slightly longer than endopodal lobe, with two lateral and two apical pinnate setae; outer proximal seta shortest, innermost longest; with some anterior spinules near inner margin.

P6 (Figure 5F) represented by single plate bearing three naked setae of which the outer is the longest, and the middle is the shortest. Copulatory pore large, crescentic, and located at slightly distal region from median line of genital double somite.

*Description of Male.* Total body length 450 μm (measured from anterior margin of rostrum to posterior margin of caudal rami). Largest width (150 μm) measured at posterior margin of cephalic shield. Body surface smooth without ornamentations (Figure 2D). Each body somites with serrated posterior margin, and few sensillae as in female. Cephalothorax slightly depressed with parallel lateral margins. Rostrum bell-shaped with pointed apical margin,

and pair of sensilla as in female (Figure 2D). P3- and P4-bearing somites slightly depressed showing lateral expansion as in Figure 2D. Pseudoperculum well developed and covering anal somite as in female. Caudal rami 1.5 times longer than anal somite, and three times longer than wide. Sexual dimorphism expressed in A1, P2, P5, P6, and segmentation of urosome.

Antennule seven-segmented (damaged during preparation, and not figured). Subchirocer with geniculation between segments 5 and 6. Mouthparts, P1, P3, and P4, as in female.

P2 endopod (Figure 5G) two-segmented, with enp-2 with hook-shaped apophysis and with 2 pinnate setae.

P5 (Figure 5H) with baseoendopod forming a shallow lobe with two small pinnate setae fused to exopod, and with pore present between baseoendopod and exopod. Exopod fused to baseoendopod forming a rectangular lobe with inner seta and two apical pinnate setae. Single surface seta isolated along outer side.

P6 (not figured) represented on both sides by ventral plate close to posterior margin of somite; each plate bearing one naked and two apical spines.

*Remarks*. Based on the keys to species of *Zosime* [8,14], *Z. montagnai* sp. nov. is morphologically similar to *Z. paratypica* Becker and Schriever, 1979. These species share the morphological characters of female A1 seven-segmented P1 enp-2 with four setae, P3 enp distal segment with four setae, P2–P4 exp-3 with seven setae, and length:width ratio of caudal rami. However, *Z. montagnai* sp. nov. and *Z. paratypica* can be distinguished from each other by the combination of the following morphological characteristics. *Z. montagnai* sp. nov. has less setae on A2 exp-3 (three setae in the new species), P4 enp distal segment, and female P5. *Z. paratypica* has one more seta, respectively. In addition, the length:width ratio of the second antennulary segment in the female of *Z. montagnai* sp. nov. is bigger than in *Z. paratypica*. Considering most of the morphological differences in terms of oligomerization, *Z. montagnai* sp. nov. is deemed to be comparatively more derived than *Z. paratypica*.

*Zosime thistlei* sp. nov.

(Figures 6–8)

*ZooBank Registration LSID*

urn:lsid:zoobank.org:act:CCEA3D8C-0B0C-4E27-951A-6A75514BDAA1

*Type Locality*. IXN500 station (20°0′32″ N, 92°23′11.4″ W) in the southern Gulf of Mexico, north-west Atlantic Ocean (depth: 1240 m).

*Material Examined.* Holotype: 1♀(MABIK CR00249459), Paratype: 1♂(MABIK CR00249460) from IXW500.

*Etymology.* The species was named in honor to Dr. David Thistle (Florida State University) who first introduced the world of Copepods to Melissa Rohal Lupher. It is a noun in the genitive case, gender masculine.

*Differential Diagnosis*. Rostrum bell-shaped, with round apical margin and A1 seven-segmented in both sexes. Caudal rami slightly longer than wide. Female P5 with seven setae, without seta on exp surface. Male P2 endopod three-segmented, enp-1 and enp-2 incompletely separated, each with inner pinnate seta, and enp-3 with hook-shaped apophysis and with one pinnate seta.

*Description of Female*. Total body length of holotype 328 µm (measured from anterior margin of rostrum to posterior margin of caudal rami). Largest width (184 µm) measured at posterior margin of cephalic shield. Urosome distinctly narrower than prosome (Figure 6A).

Cephalothorax trapezoid with serrate posterior margin; dorsal surface smooth with few sensilla posteriorly. Prosomites (Figure 6A) with smooth dorsal surface, serrated posterior margins and few sensilla posteriorly as figured. Each somite with pointed lateral posterior end.

Rostrum fused to cephalothorax, with rounded and shallow apical margin, bell-shaped, and round at apical margin and bearing two sensilla (Figure 6A).

Urosomites (Figure 6A) with serrated posterior margin as illustrated. Dorsal surface and posterior margin of P5 bearing somite ornamented as preceding somites. Genital double-somite with smooth dorsal surface. Posterior margins of each urosomite finely serrated. Each urosomite with several posterior sensilla as illustrated.

Anterior half of anal somite (Figure 6A) covered with well-developed pseudoperculum, unornamented except for few spinules along lateral distal corner and median distal margin.

Caudal rami (Figures 6A and 7C) as long as anal somite, and slightly longer than wide; setae I and II arising laterally halfway along outer margin, seta I smallest; seta III as long as seta II; seta IV basally fused to seta V; seta V longest and slightly longer than all urosomites combined; seta VI shorter than seta III and located on distal inner corner; seta VII bare, located dorsally, triarticulated.

Antennule (Figure 7A) seven-segmented. Segment 1 with row of strong spinules around inner margin. Segment 2 longest. Segment 3 with ae fused basally to strong pinnate seta. All setae pinnate except for four, one, two, and four naked setae on second, third, sixth, and last segments, respectively. Armature formula: 1-(1 pinnate), 2-(4+6 pinnate), 3-(2 + 1 pinnate+(1+ae)), 4-(2 pinnate), 5-(1), 6-(2), 7-(3+1 acrothek). Apical acrothek consisting of well-developed but small ae fused basally to strong and stout pinnate seta, and short naked seta.

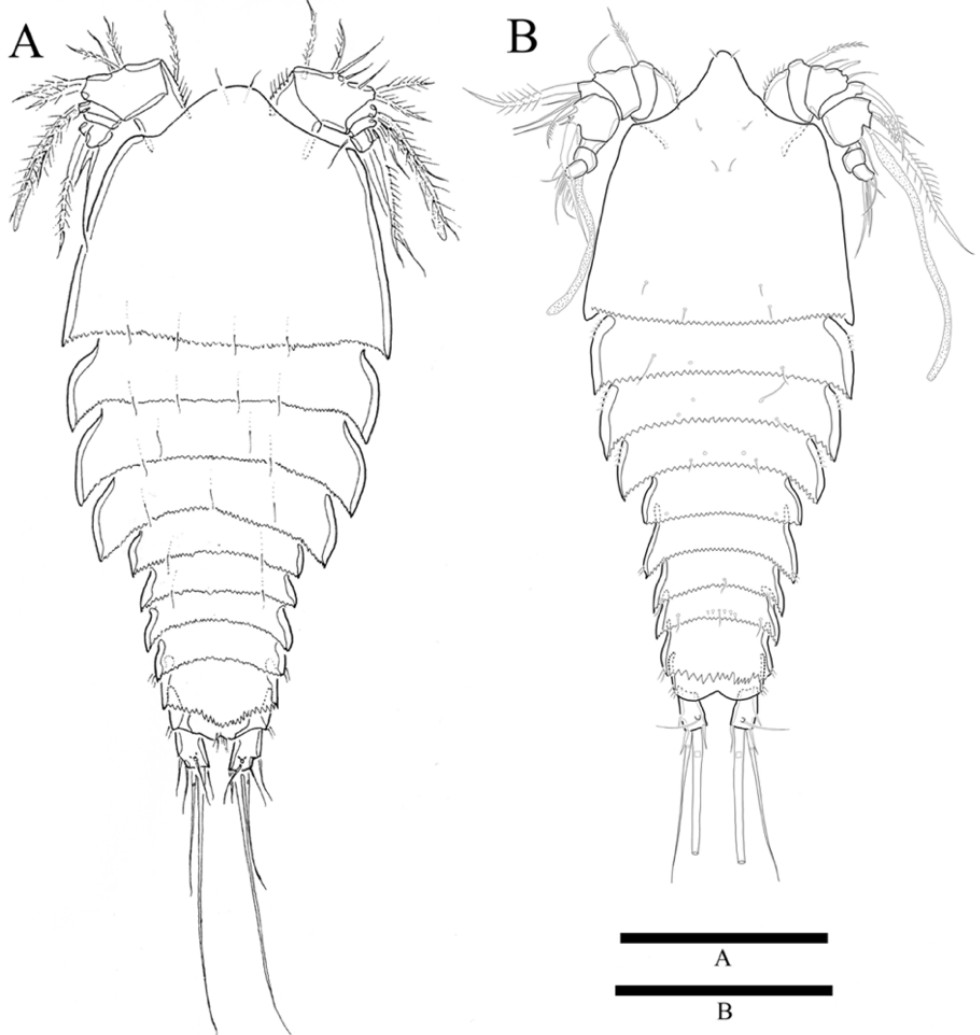

**Figure 6.** *Zosime thistlei* sp. nov. (**A**) Habitus of female, dorsal; (**B**) Habitus of male, dorsal. Scale bars: 100 µm.

Antenna (Figure 7B) four-segmented, comprising coxa (not figured), basis, and free two-segmented endopod. Basis with row of long inner spinules, and with subdistal abexopodal seta. Endopodal segment-1 naked with no surface ornamentations; enp-2 with row of inner spinules proximally; apical armature consisting of stout pinnate spine and three geniculate spines, and strong pinnate innermost spine basally fused to laterally with short pinnate seta; short pinnate spine and two pinnate stout spines. Exopod three-segmented, with one, one, and four setae, respectively; exp-1 and exp-2 with pinnate spines; second segment shortest; first segment twice as long as second; last segment much longer than preceding two segments combined, with lateral spine and three apical pinnate spines.

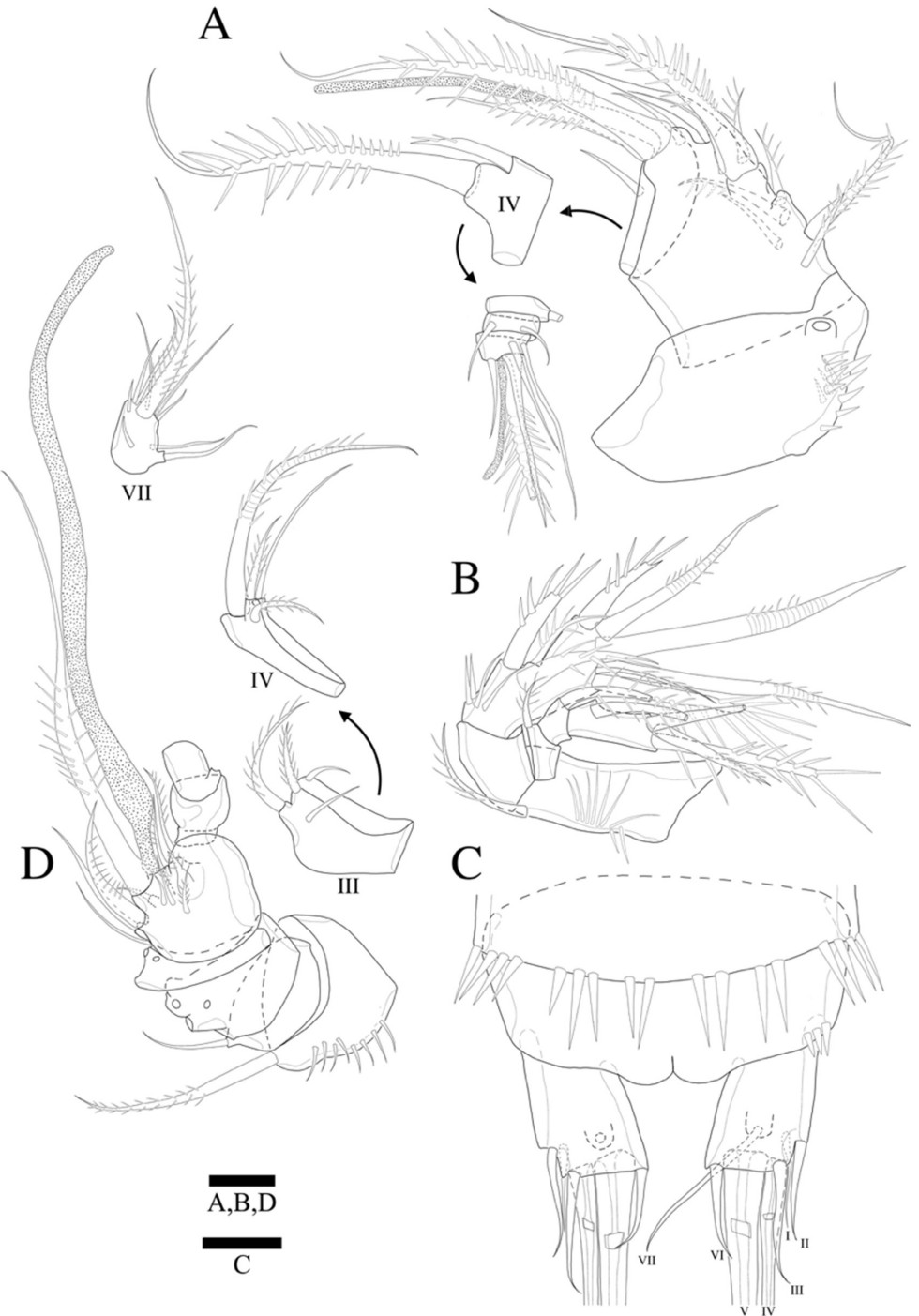

**Figure 7.** *Zosime thistlei* sp. nov. Female. (**A**) A1; (**B**) A2; (**C**) Anal somite and caudal rami. Male. (**D**) A1. Scale bars: 10 μm.

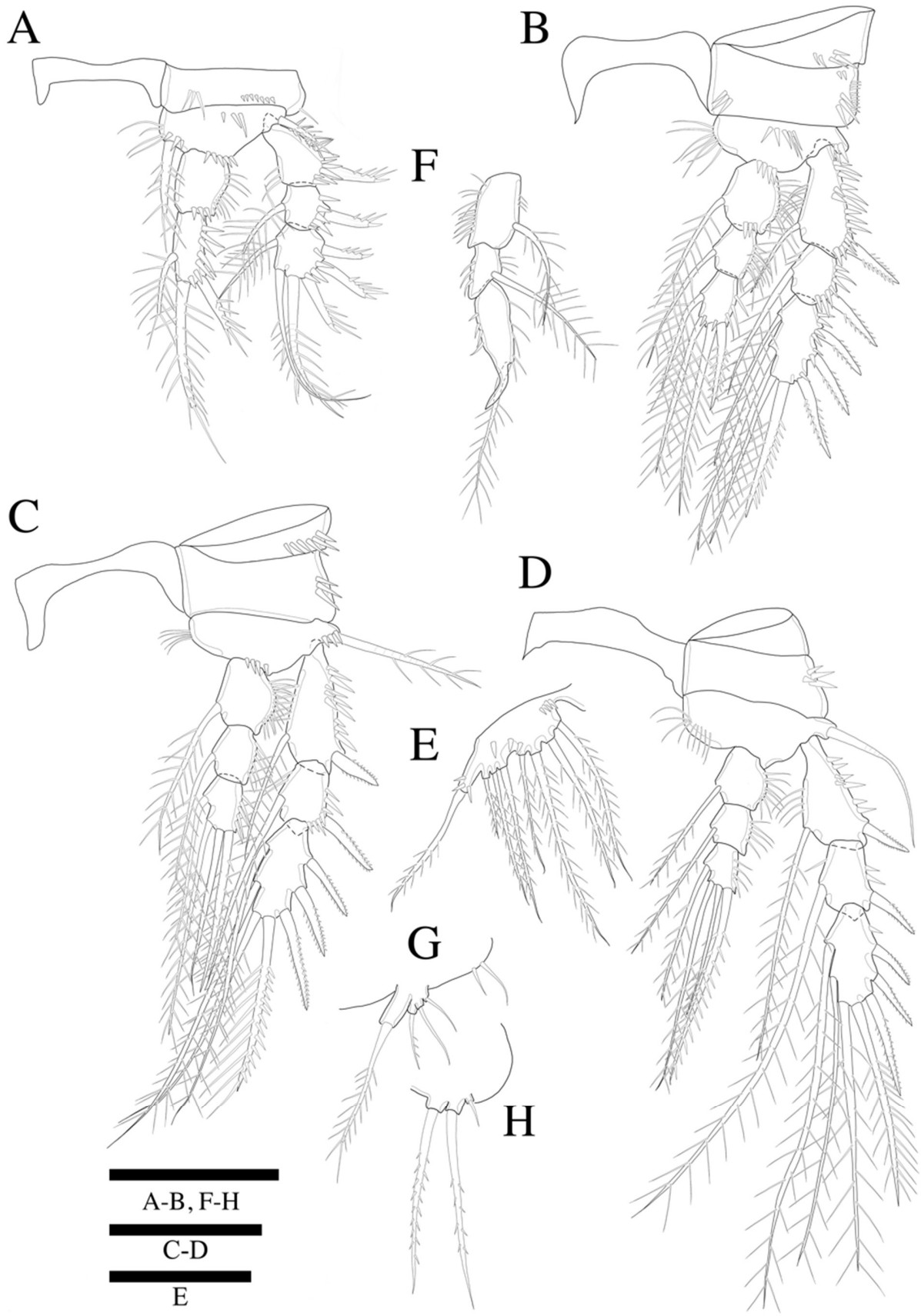

**Figure 8.** *Zosime thistlei* sp. nov. Female. (**A**) P1; (**B**) P2; (**C**) P3; (**D**) P4; (**E**) P5. Male. (**F**) P2 endopod; (**G**) P5; (**H**) P6. Scale bars: 50 μm.

Mandible, maxillule, maxilla, maxilliped as in *Z. montagnai* with no significant discrepancies.

Swimming legs 1–4 (Figure 8A–D) biramous, with three-segmented exopods, P1 endopod two-segmented, P2–P4 endopods three-segmented. Coxae and bases with anterior rows of surface spinules as illustrated. Praecoxae triangular. Coxa rectangular.

P1 (Figure 8A). Basis with strong pinnate inner seta and with spinules along inner margin, and with pinnate outer seta and several spinules along outer margin; with row of spinules near endopod, and on median anterior surface. Exopod three-segmented; exp-1 and exp-2 with pinnate outer spine, respectively; exp-2 with inner pinnate seta; exp-3 with three pinnate outer spines, two pinnate distal setae and inner pinnate seta. Endopod two-segmented; enp-1 unarmed and ornamented as shown, enp-2 slightly longer than enp-1, and with inner seta and two apical spines of which the inner apical spine is two times longer than the outer one.

P2 (Figure 8B). Basis with row of long inner spinules, and with seta (damaged, not figured) and several outer spinules. Exopod longer than endopod; exp-1 longest and exp-2 shortest; exp-1 and exp-2 with row of inner spinules. Endopod three-segmented; enp-1 longest, enp-3 slightly longer than enp-2 in length; each segment with row of outer spinules; enp-3 reaching only middle of exp-3.

P3 (Figure 8C). Basis with patch of long inner spinules, and with few spinules at base of pinnate outer seta. Exopod longer than endopod; exp-1 longest and exp-2 shortest; exp-1, exp-2 with row of inner setules. Endopod only reaching to proximal 1/3 of exp-3; each endopodal segment with row of long outer spinules; enp-2 and enp-3 subequal in length.

P4 (Figure 8D). Basis with row of long inner spinules, and with bare outer seta. Exopod longer than endopod; exp-1 and exp-2 with row of inner spinules; exp-1 longest, and exp-2 shortest. Each endopodal segments with row of outer spinules; enp-1 slightly longer than other two segments; enp-2 and enp-3 subequal in length; endopod not reaching to distal margin of exp-2. Armature formula as in Table 2.

**Table 2.** Armature formulae of legs 1–4.

|  | Exopod | Endopod |
|---|---|---|
| P1 | 0.1.123 | 0.111 |
| P2 | 1.1.223 | 1.1.121 |
| P3 | 1.1.223 | 1.1.121 |
| P4 | 1.1.223 | 1.1.121 |

P5 (Figure 8E). Outer basal seta long and pinnate, set on cylindrical setophore bearing few spinules on anterior surface. Exopod fused to baseoendopod. Endopodal lobe forming a shallow lobe with four long pinnate setae, outermost one shortest, outer middle one longest, and the other two inner ones subequal in length; with rows of outer spinules anteriorly. Exopod also forming a shallow lobe with three pinnate setae, of which the outer is the shortest, and the innermost is the longest.

P6 damaged (not figured).

*Description of Male*. Total body length 329 μm (measured from anterior margin of rostrum to posterior margin of caudal rami). Largest width (132 μm) measured at posterior margin of cephalic shield. Body surface smooth, without ornamentations. Each body somites with serrated posterior margin, and few sensillae as in female. Cephalothorax as long as wide including rostrum. Rostrum more prominent than in female, bell-shaped with round apical margin, and pair of sensilla as in female (Figure 6B). Pseudoperculum well-developed and covering proximal half of anal somite as in female. Caudal rami as long as anal somite, and slightly longer than wide. Sexual dimorphism expressed in A1, P2, P5, P6, and segmentation of urosome.

Antennule (Figure 7D) seven-segmented. Subchirocer, with geniculation between segments 5 and 6. Segment 1 with row of strong spinules around inner margin. Segment 5 swollen and longest with ae fused basally to strong pinnate seta. Armature formula: 1-(1 pinnate), 2-(1), 3-(2 + 2 pinnate), 4-(1+3 pinnate), 5-(4 + 6 pinnate + (1 pinnate+ae)),

6-(2 irregular processes), 7-(7+1 acrothek). Apical acrothek consisting of well-developed but small ae fused basally to strong pinnate spine, and naked seta.

Mouthparts, P1, P3, and P4 (not shown), as in female.

P2 endopod (Figure 8F) three-segmented, with rows of spinules along inner and outer margins; enp-1 with pinnate seta; enp-2 shortest with pinnate seta; enp-3 modified, longest, with hook-shaped apophysis and pinnate seta.

P5 (Figure 8G). Baseoendopod forming a shallow lobe fused to an exopod, with two small setae. Exopod forming a small lobe with pinnate outer and two naked inner apical setae. Basal seta long and pinnate set on cylindrical setophore.

P6 (Figure 8H), vestigial, plate bearing short naked seta and two long pinnate apical setae.

*Remarks*. Based on keys to the species of *Zosime* [8,14], *Z. thistlei* sp. nov. is morphologically similar to *Z. atlantica* Bodin, 1968. *Z. thistlei* sp. nov. and *Z. atlantica* share the characters of female A1 seven-segmented, P1 enp-2 with three setae, P4 enp-3 with four setae, P2–P4 exp-3 with seven setae. However, *Z. thistlei* sp. nov. and *Z. atlantica* can be distinguished from each other by the combination of the following morphological characteristics: (1) *Z. thistlei* sp. nov. has less setae on female P5. *Z. atlantica* has one more seta, (2) The cleft between the baseoendopod and exopod of the female P5 is less pronounced in *Z. thistlei* than in *Z. atlantica*, (3) armature of the A2 exp-3 (with seta in *Z. thistlei*, unarmed in *Z. atlantica*), (4) length of distal segments of P4 enp and exp are similar with mid-segments, but, those of *Z. atlantica* are longer than mid-segments, (5) length/width ratio of the caudal rami (2 in *Z. thistlei*, but 3 in *Z. atlantica*).

*Zosime tunnelli* sp. nov.
(Figures 9 and 10)

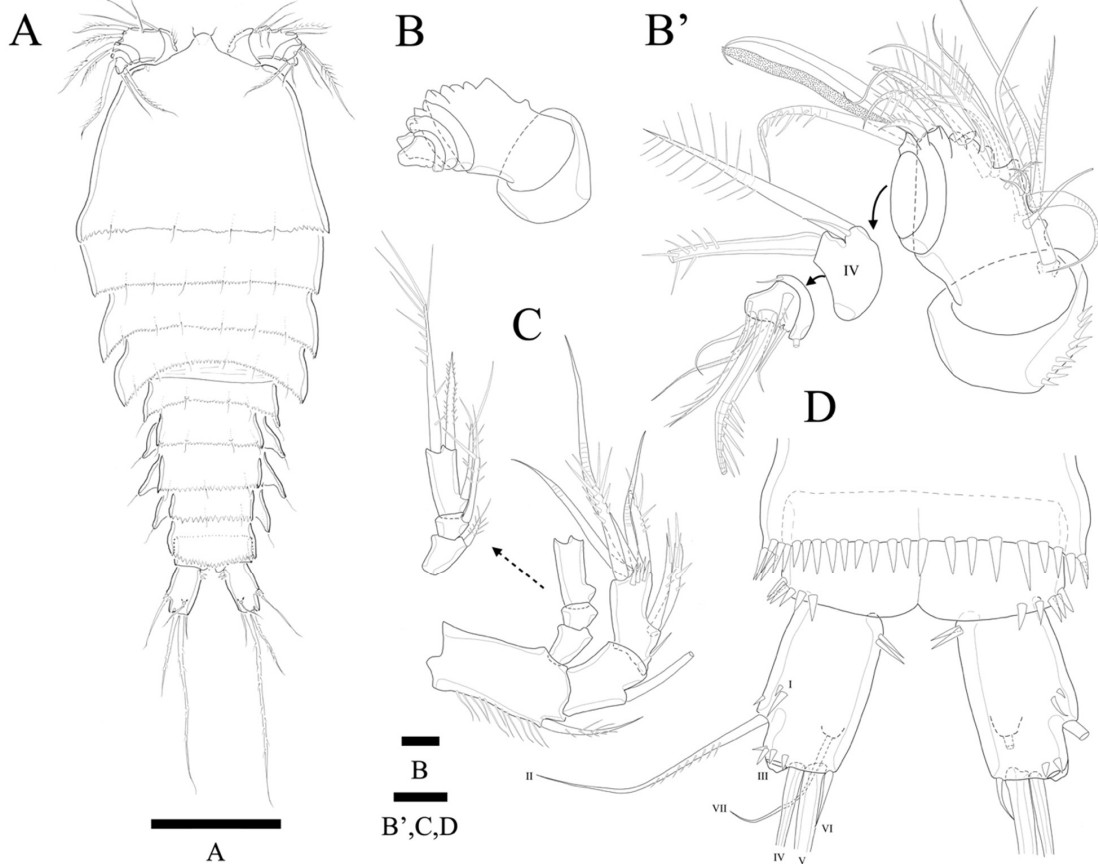

**Figure 9.** *Zosime tunnelli* sp. nov. (**A**) Habitus, dorsal; (**B**) A1 showing segmentation, armature omitted (**B'**) A1; (**C**) A2; (**D**) Anal somite and caudal rami. Scale bars: (**A**) 100 µm; (**B,B'–D**) 10 µm.

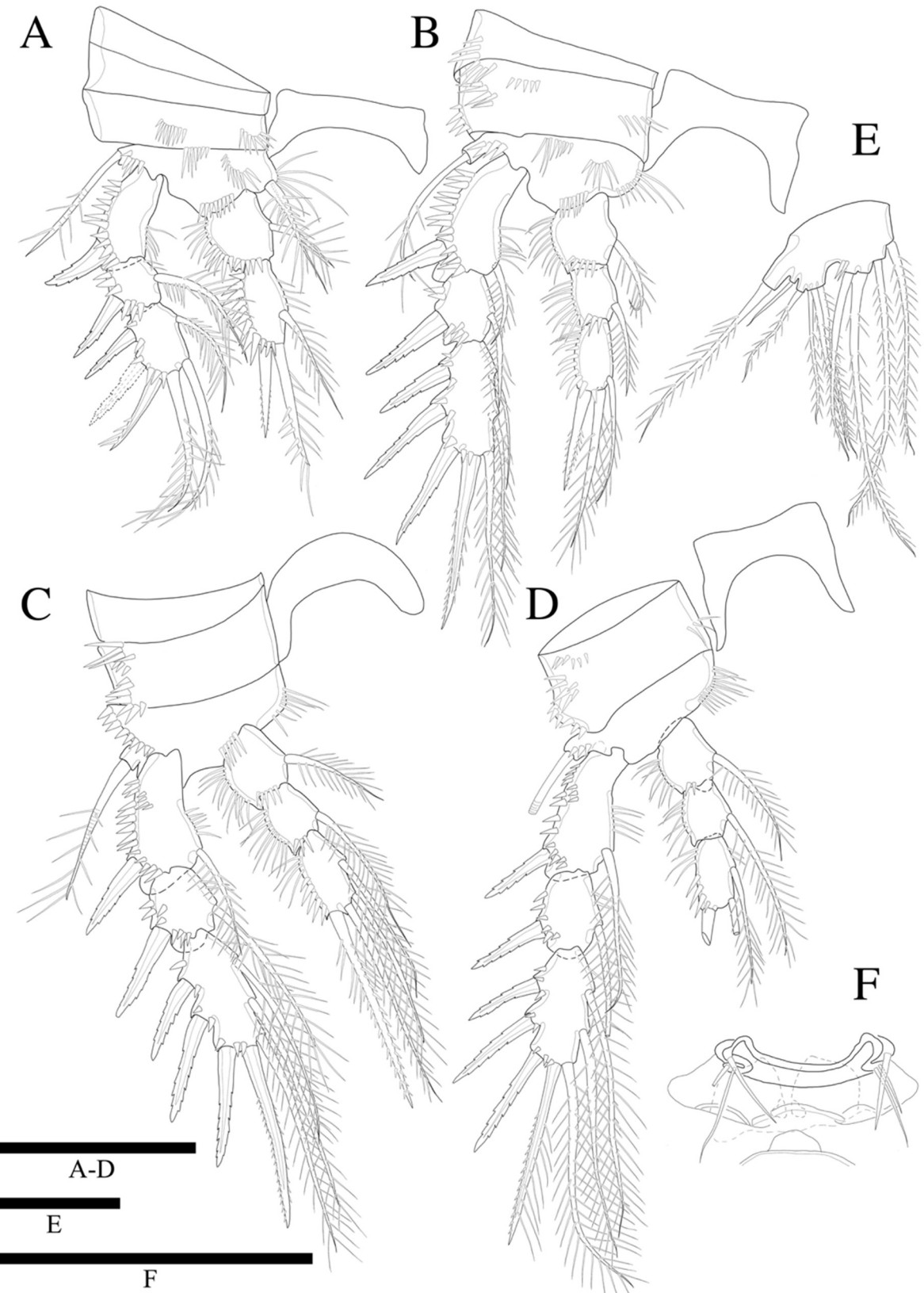

**Figure 10.** *Zosime tunnelli* sp. nov. (**A**) P1; (**B**) P2; (**C**) P3; (**D**) P4; (**E**) P5. (**F**) P6 endopod. Scale bars: 50 μm.

*Type Locality*. IXW100 station (19°25′7.32″ N, 92°41′21.8″ W) in the southern Gulf of Mexico, north-west Atlantic Ocean (depth: 179 m).

*Material Examined.* Holotype: 1♀(MABIK CR00249461)

*Etymology*. The species is named in honor to Prof. John Wesley Tunnell Jr. (Harte Research Institute for Gulf of Mexico Studies, Texas A&M University—Corpus Christi) in recognition of his accomplishments and contributions to the ecological knowledge of the Gulf of Mexico. It is a noun in the genitive case, gender masculine.

*Differential Diagnosis*. Based on female. Body slightly flattened. Prosome 1.4 times longer than urosome, and distinctly wider than urosome Rostrum bell-shaped, and smoothly round apically. Antennule six-segmented. Genital somite and succeeding urosomal segments with lateral expansions. Caudal rami 1.4 times longer than anal somite, and 1.5 times longer than wide. P5 with nine setae, without seta on exp surface.

*Description of Female*. Total body length of holotype 452 μm (measured from anterior margin of rostrum to posterior margin of caudal rami). Largest width (223 μm) measured at posterior margin of cephalic shield (Figure 9A). Body slightly flattened. Prosome 1.4 times longer than urosome, and distinctly wider than urosome.

Cephalothorax 1.3 times shorter than length, with serrate posterior margin; dorsal surface smooth with few sensilla posteriorly. Prosomites (Figure 9A) with smooth dorsal surface, serrated posterior margins and few sensilla posteriorly as figured.

Rostrum fused to cephalothorax, bell-shaped, smoothly round apically, and with two sensilla (Figure 9A).

Urosomites (Figure 9A) with serrated posterior margin as illustrated. Dorsal surface and posterior margin of P5 bearing somite ornamented as preceding somites. Genital double-somite with smooth dorsal surface. Posterior margins of each urosomite serrated with rough denticles. Genital somite and succeeding urosomal segments produced laterally. Each urosomite with several posterior sensilla as illustrated.

Anal somite (Figure 9A) totally covered with well-developed pseudoperculum, unornamented.

Caudal rami (Figure 9A,D) 1.4 times longer than anal somite, and 1.5 times longer than wide; setae I and II arising laterally halfway along outer margin, seta II longer than seta I; seta III pinnate and much longer than setae II; seta IV and seta V pinnate; seta V longest, but much shorter than all urosomites combined; seta VI located on distal inner corner; seta VII bare, located dorsally, triarticulated.

Antennule (Figure 9B,B') six-segmented. Segment 1 with row of strong spinules around inner margin. Segment 2 largest. Segment 3 with ae fused basally to strong pinnate seta. All setae pinnate except for nine, one, two, and four naked setae on second, fourth, fifth, and last segments, respectively. Armature formula: 1-(1 pinnate), 2-(9+5 pinnate), 3-(1 pinnate+(1+ae)), 4-(1+2 pinnate), 5-(2), 6-(3+2 pinnate+1 acrothek). Apical acrothek consisting of well-developed but small ae fused basally to strong and stout pinnate seta, and short seta.

Antenna (Figure 9C) four-segmented, comprising coxa, basis, and free two-segmented endopod. Coxa small without ornamentations. Basis with row of spinules along inner margin with pinnate abexopodal seta. First endopodal segment as long as second segment with abexopodal seta; segment 2 with row of stout spinules on apical margin and row of spinules around inner lateral margin; apical armature consisting of two stout pinnate and three geniculate spines; pinnate apical spine basally fused to naked small seta; two pinnate spines laterally. Exopod three-segmented, with one, one, and three setae, respectively; all setae and spines pinnate; second segment shortest; first segment twice as long as second; last segment much longer than preceding two segments combined with lateral seta and two apical pinnate setae.

Mandible, maxillule, maxilla, and maxilliped are the same as in *Z. montagnai* sp. nov. without significant discrepancies.

Swimming legs 1–4 (Figure 10A–D) biramous, with three-segmented exopods, and three-segmented endopod in P2–P4, except for P1 endopod having two-segmented endopod, and

with wide intercoxal sclerites and well-developed praecoxae. Coxae and bases with anterior rows of surface spinules as illustrated. Praecoxa narrow sclerite. Coxa rectangular.

P1 (Figure 10A). Basis with strong inner pinnate seta, and with spinules along inner margin, and with outer pinnate seta, and several spinules along outer margin; several rows of spinules on anterior surface. Exopod three-segmented; exp-1 and exp-2 with pinnate outer spine, respectively; exp-2 with inner pinnate seta; exp-3 with three pinnate outer spines (one spine missing), two pinnate distal, and inner pinnate seta. Endopod two-segmented; enp-2 slightly longer than enp-1, and with inner seta and two apical spines; inner apical spine about 1.4 times longer than outer one.

P2 (Figure 10B). Basis with row of long spinules along inner distal margin, and with outer pinnate seta, and several outer spinules along outer margin; rows of spinules on anterior surface. Exopod longer than endopod; exp-1 longest, exp-2 shortest; exp-1 with row of inner spinules. Endopod three-segmented; enp-1 and enp-3 subequal in length, enp-2 shortest; each segment with row of spinules along outer lateral margin; enp-3 reaching only half of exp-3.

P3 (Figure 10C). Basis with row of long spinules along inner distal margin, and with outer pinnate long spine, and row of outer spinules. Exopod much longer than endopod; exp-1 and exp-3 subequal in length, exp-2 shortest; each segment with row of inner setules. Each endopod segment with row of long spinules along outer lateral margin; enp-1 slightly longer than enp2; enp-3 longest; enp-3 only reaching to proximal 1/3 of exp-3.

P4 (Figure 10D). Basis with row of long spinules along inner distal margin, and with outer seta, and several spinules along outer margin. Exopod much longer than endopod; all exopod segments with row of spinules along inner lateral margin; exp-1 longest, and exp-2 shortest. Each endopod segments with row of spinules along outer lateral margins; enp-1 and enp-2 subequal in length; enp-3 longest; enp-3 only reached to middle of exp-2. Armature formulae as in Table 3.

**Table 3.** Armature formulae of legs 1–4.

|     | Exopod  | Endopod |
| --- | ------- | ------- |
| P1  | 0.1.123 | 1.111   |
| P2  | 1.1.223 | 1.1.111 |
| P3  | 1.1.223 | 1.1.121 |
| P4  | 1.1.223 | 1.1.120 |

P5 (Figure 10E). Baseoendopod forming a shallow lobe with four long pinnate setae and fused to exopod. Exopod rectangular shape with outer seta and three apical setae, and isolate naked seta located between exopod and basal cylindrical process. Basal seta long and pinnate arising from elongated cylindrical process.

P6 (Figure 10F) represented by single plate bearing three naked setae; middle shortest.
*Male.* Unknown.

*Remarks.* *Z. tunnelli* sp. nov. is morphologically similar to *Z. destituta* Kim J.G., Jung and Yoon, 2016. These species share the morphological characters of P1 enp-2 with three setae, P3 enp distal segment with four setae, P4 enp distal segment with three setae, P2–P4 exp-3 with seven setae. However, *Z. tunnelli* sp. nov. and *Z. destituta* can be distinguished from each other by the combination of the following morphological characteristics: (1) *Z. tunnelli* sp. nov. has less setae on A2 exp-3 (one more in *Z. destituta*); (2) *Z. tunnelli* sp. nov. has one more seta on female P5 than *Z. destituta*; (3) distance of female P5 baseoendopod and exp is closer in *Z. tunnelli* sp. nov.; (4) P6 setae of *Z. tunnelli* sp. nov. are all long and naked, while one of them is relatively short and pinnate in *Z. destituta*; (5) location of inner seta on P4 enp-3 is more distal in *Z. destituta*; (6) seta type and length of caudal rami seta II is shorter and naked in *Z. destituta,* while the seta II is longer and pinnate in *Z. tunnelli* sp. nov.

Genus *Peresime* Dinet, 1974
Type species
*Peresime abyssalis* Dinet, 1974

*Peresime pryorae* sp. nov.
(Figures 11 and 12)

**Figure 11.** *Peresime pryorae* sp. nov. (**A**) Habitus, dorsal; (**B**) A1; (**C**) A2; (**D**) Mandible; (**E**) Maxillule; (**F**) Maxilla; (**G**) Maxilliped. Scale bars: (**A**) 100 μm; (**B–G**) 10 μm.

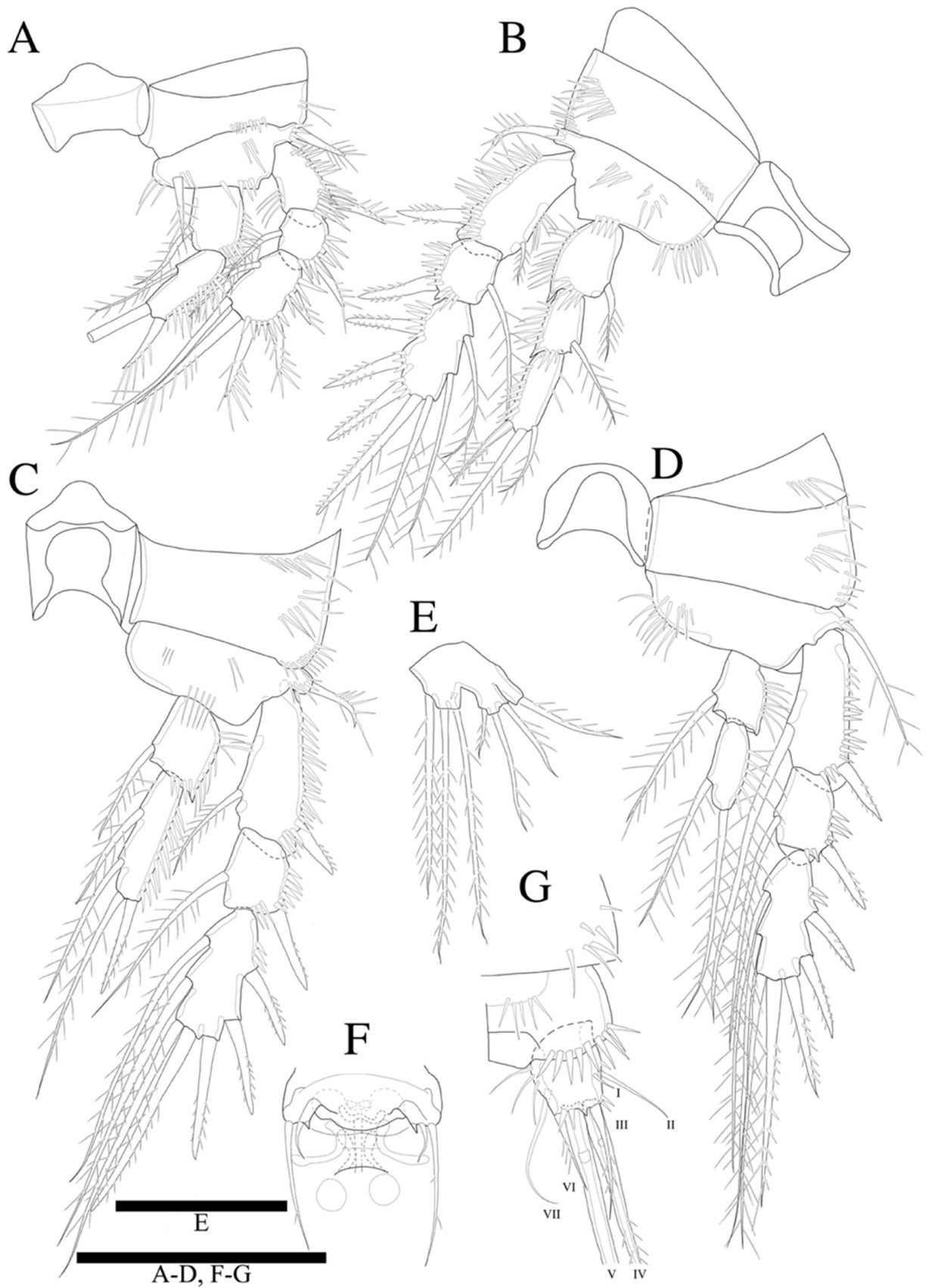

**Figure 12.** *Peresime pryorae* sp. nov. (**A**) P1; (**B**) P2; (**C**) P3; (**D**) P4; (**E**) P5; (**F**) P6; (**G**) Caudal ramus. Scale bars: 50 μm.

*ZooBank Registration LSID*
urn:lsid:zoobank.org:act:05DA35B7-BA36-4F3F-AD35-F87C7EF46EFF

*Type Locality*. IXW500 station (19°26′39.1″ N, 93°53′19.3″ W) in the southern Gulf of Mexico, north-west Atlantic Ocean (depth: 1,010 m).

*Materials examined.* Holotype: 1♀(MABIK CR00249462)

*Etymology.* The specific name refers to Marissa Pryor (Harte Research Institute for Gulf of Mexico Studies, Texas A&M University—Corpus Christi) in recognition of her hard work and dedication to the study of meiofauna during her time as an undergraduate assistant in helping Melissa Rohal Lupher complete her PhD lab work. it is a noun in the genitive case, gender feminine.

*Differential diagnosis*. Based on female. Rostrum triangular. Each somite armed with smooth posterior margin. Antennule eight-segmented. Caudal rami slightly longer than wide. P2 endopod three-segmented. P1, P3, and P4 with two-segmented endopods. P5 exopod fused to baseoendopod, with three setae.

*Description of female.* Total body length of holotype 376 μm (measured from anterior margin of rostrum to posterior margin of caudal rami). Largest width (126 μm) measured at posterior margin of cephalic shield (Figure 11A). Urosome distinctly narrower than prosome.

Cephalothorax somewhat triangular with smooth posterior margin; dorsal surface smooth with few sensilla posteriorly. Prosomites (Figure 11A) with smooth dorsal surface, row of spinules along posterior margins and few sensilla posteriorly as figured.

Rostrum fused to cephalothorax, triangular, and with round apical margin and bearing two sensilla (Figure 11A,B).

Anal somite (Figure 11A) completely covered with well-developed operculum with serrated distal margin.

Urosomites (Figure 11A) with few sensilla, and row of posterior spinules as illustrated. Dorsal surface and posterior margin of P5 bearing somite ornamented as preceding somites. Genital double-somite with smooth dorsal surface. Succeeding urosomal somite after genital somite, with smooth posterior margin and few spinules along both lateral posterior corner; posterior margins of all other urosomites deeply serrate and with several sensilla as illustrated.

Caudal rami (Figures 11A and 12G) slightly longer than anal somite, and 1.1 times longer than wide; setae I and II arising laterally along outer margin, seta II longer than seta I; seta III longer than setae II; seta IV basally fused to seta V; seta V longest, much longer than all urosomites combined; seta VI shorter than seta III and located on distal inner corner; seta VII bare, located dorsally, triarticulated.

Antennule (Figure 11B) eight-segmented. Segment 1 with row of strong spinules around proximal margin. Segments 2 and 3 subequal in length. Segment 4 with ae fused basally to strong pinnate seta. All setae pinnate except for two, two, one, one naked seta on second, third, fifth and sixth segments, respectively. Armature formula: 1-(1), 2-(2 + 3 pinnate), 3-(2 + 5 pinnate), 4-(1 pinnate + (1+ae)), 5-(1 + 2 pinnate), 6-(1 + 2 pinnate), 7-(1 pinnate), 8-(3 pinnate).

Antenna (Figure 11C) four-segmented, comprising coxa, basis, and free two-segmented endopod. Basis with row of anterior spinules, with pinnate abexopodal seta. Endopodal segment 1 smooth with seta; segment 2 with row of stout apical spinules; apical armature consisting of pinnate spine, four pinnate setae, and naked seta; three pinnate spines and naked spine laterally. Exopod three-segmented, with one, one, and four setae, respectively; second segment shortest; first segment about 1.3 times longer than second; last segment much longer than preceding two segments combined with lateral seta and three apical pinnate setae.

Mandible (Figure 11D). Well-developed with about six multiple faced sharp teeth and naked seta at distal corner. Basis with pinnate seta. Endopod fused to basis forming one plate, with one lateral and four apical setae. Exopod one-segmented with lateral seta and two apical pinnate setae.

Maxillule (Figure 11E). Praecoxa without distinct spinular ornamentation. Arthrite strongly developed, with two naked surface setae, seven spines/setae around distal margin. Coxa with cylindrical endite bearing two apical setae. Basoendite with seven naked setae and pinnate seta. Exopod one-segmented, smaller than endopod with three pinnate setae.

Maxilla (Figure 11F). Syncoxa with three endites. Proximal endite bilobate; basal lobate with two pinnate setae; distal lobate with naked seta and two pinnate setae. Medial and distal endite with pinnate spine and two naked setae, and two pinnate spines, respectively. Allobasis produced into strong curved claw and overlapped curved spine with short slender seta basally; accessory armature consisting of two slender lateral setae proximally, and close to base of endopod. Endopod 1-segmented with two slender distal setae.

Maxilliped (Figure 11G). Syncoxa elongate and cylindrical with seta. Basis with row of outer spinules. Endopod small with long sparsely pinnate seta, short apical seta, and two naked lateral setae.

Swimming legs 1–4 (Figure 12A–D) biramous, with three-segmented exopods, endopods of P1, P3, and P4 two-segmented, of P2 three-segmented with wide intercoxal sclerites and well-developed praecoxae. Coxae and bases with anterior rows of surface spinules as illustrated. Praecoxa triangular. Coxa rectangular.

P1 (Figure 12A). Basis with strong pinnate outer seta, with inner spinules and several outer spinules. Exopod three-segmented; exp-1 and exp-2 with pinnate outer spine, respectively; exp-2 with inner pinnate seta; exp-3 with three pinnate outer spines, two pinnate distal, and inner pinnate seta. Endopod two-segmented; enp-2 slightly longer than enp-1, and with inner seta and two apical spines; inner apical spine clearly longer than outer one (broken during preparation).

P2 (Figure 12B). Basis with row of long inner spinules distally, and with pinnate outer seta, and rows of outer spinules anteriorly. Exopod and endopod subequal in length; exp-1 longest and exp-2 shortest; exp-3 with row of inner spinules; exp-1 with small pinnate inner seta, Endopod three-segmented; enp-3 longest, enp-1 and enp-2 subequal in length; each segment with row of outer spinules; enp-3 reaching to distal margin of exp-3.

P3 (Figure 12C). Basis with row of spinules distally close to insertion site of the enp, and with pinnate relatively short spine and few scattered spinules at the base of the outer seta. Exopod longer than endopod; exp-1 longest, exp-2 shortest; exp-1 with row of inner setules. Endopod only reaching to proximal 1/3 of exp-3; each endopodal segment with row of long outer spinules; enp-2 1.6 times longer than enp-1; enp-1 with inner distal pinnate seta; enp-2 with two inner pinnate setae and distal elements as shown.

P4 (Figure 12D). Basis with rows of long inner spinules distally on anterior surface, with pinnate outer seta and several outer spinules. Exopod much longer than endopod; exp-1 longest, and exp-2 shortest; exp-1 and exp-2 with pinnate outer spine and long inner pinnate seta, respectively; exp-3 with two inner pinnate setae and two distal pinnate setae, and with three outer spines. Each endopodal segments with row of outer spinules; enp-1 with inner pinnate seta; enp-2 clearly longer than enp-1, with two pinnate setae; endopod reaching to middle region of exp-2. Armature formulae as in Table 4.

**Table 4.** Armature formulae of legs 1–4.

|     | Exopod | Endopod |
| --- | --- | --- |
| P1 | 0.1.123 | 1.120 |
| P2 | 1.1.223 | 1.1.111 |
| P3 | 1.1.223 | 1.220 |
| P4 | 1.1.223 | 1.110 |

P5 (Figure 12E). Outer basal seta long and pinnate set on cylindrical setophore. Exopod fused to baseoendopod. Endopodal lobe trapezoid with three long pinnate apical setae; innermost one shortest, other two subequal in length; with few scattered inner spinules anteriorly. Exopod rectangular, slightly higher than the endopodal lobe, with three apical pinnate setae; inner seta shortest, and middle apical seta longest.

P6 (Figure 12F) represented by single plate bearing three elements; outer is the longest, inner is the shortest, small denticle shaped spine. Copulatory pore large, crescentic, and located at slightly distal region from median line of genital double somite.

*Male.* Unknown.

*Remarks.* Based on Well's (2007) key to the species of Zosimeidae [14], *P. pryorae* sp. nov. is significantly similar to *P. reducta* (Becker and Schriever, 1979). They share the setal formulae of P1–P5, length:width ratio of caudal rami, and length of the body. However, *P. pryorae* sp. nov. and *P. reducta* can be distinguished from each other by the combination of the following morphological characteristics: (1) *P. pryorae* sp. nov. has less setae on mandibular palp; *P. reducta* has one more seta and all setae are naked (four setae in *P. reducta*), (2) *P. pryorae* sp. nov. has more setae on the coxa and basoendite of the maxillule (nine setae in *P. pryorae* sp. nov. but seven setae in *P. reducta*), (3) *P. pryorae* sp. nov. has more setae on distal lobate of maxilla (three setae in *P. pryorae* sp. nov. but two setae in *P. reducta*), and maxilliped (five setae in *P. pryorae* sp. nov. but three setae in *P. reducta*), (4) *P. pryorae* sp. nov. has more segments in female A1, and (5) the length of setae of P2 exp-1 and enp-3 are shorter in *P. pryorae* sp. nov.

### 3.2. Key of Zosimeidae

Kim et al. [8] proposed three groups of species in *Zosime* according to morphological characteristics. However, Pointner [15] refuted the defining the group by the number of A1 segments. In addition, species that do not meet the existing criteria have also been reported (e.g., *Z. carsteni* Pointner, 2017 and *Z. eliasi* Pointner, 2017).

Referring to Wells [14], a new classification key of the family Zosimeidae was prepared with 27 species including the new species described here. The three species-groups of *Zosime* were judged to be controversial, so the classification key did not include any related content.

#### 3.2.1. Key to the Genera of Zosimeidae

1. P2 enp 3-segmented .................................................................................2
   P2 enp 2-segmented .................................................................................4
2. P3–P4 enp 3-segmented ...........................................................................3
   P3–P4 enp 2-segmented ...........................................................*Peresime* Dinet, 1974
3. P1 exp 3-segmented ...............................................................*Zosime* Boeck, 1873
   P1 exp 2-segmented .......................*Heterozosime tenuis* Kim, JG and J Lee, 2021
4. P2 exp-3 with 7 setae ...............................*Pseudozosime browni* Scott T., 1912
   P2 exp-3 with 5 setae ....................*Acritozosime spinesco* Kim, JG and J Lee, 2021

#### 3.2.2. Key to the Species of Peresime

1. P1 enp-2 with 3 setae .............................................................................2
   P1 enp-2 with 2 setae ...........................................*P. abyssalis* Dinet, 1974
2. Mandibular palp exp with 4 naked setae; Mxp enp with 3 setae; Female A1 6 segmented ........................................*P. reducta* (Becker and Schriever, 1979)
   Mandibular palp exp with 3 pinnate setae; Mxp enp with 5 setae; Female A1 8 segmented ...............................................................*P. pryorae* sp. nov.

#### 3.2.3. Key to the Species of Zosime

1. P2 exp-3 with 6 setae .............................................................................2
   P2 exp-3 with 7 setae .............................................................................5
2. P3 enp distal segment with 3 setae ..............................................................3
   P3 enp distal segment with 4 setae ..............................................................4
3. Caudal rami length/width ratio about 1.5; female P5 with 3 setae ...*Z. bathyalis* Por, 1967
   Caudal rami length/width ratio about 2.5; female P5 with 4 setae ...*Z. gymnokosmosa* Kim J.G. and Lee J., 2019
4. P4 enp distal segment with 3 setae ..................*Z. changi* Kim J.G. and Lee J., 2019

         P4 enp distal segment with 2 setae . . . . . . . . . . . . . . . . . *Z. comata* Kim J.G. and Lee J., 2019

5.    P3–P4 exp-3 with 8 setae . . . . . . . . . .**.**. . . . . . . . . . . . . . . . . . . . . . . . . . . . . . . . . . . . . . . . . . . . . . . . . . . 6

         P3–P4 exp-3 with 7 setae . . . . . . . . . .**.**. . . . . . . . . . . . . . . . . . . . . . . . . . . . . . . . . . . . . . . . . . . . . . . . . . . 7

6.    P3 enp distal segment with 5 setae; P1 enp-2 with 4 setae . . . . . . . . . . . . . . . . . . . . *Z. incrassata* Sars G.O., 1910 [Subspecies: *Z. incrassata bathybia* Bodin, 1968/*Z. incrassata incrassata* Sars G.O., 1910]

         P3 enp distal segment with 4 setae; P1 enp-2 with 2 setae . . . . . . . . . *Z. reyssi* Dinet, 1974

7.    P4 enp distal segment with 4 setae . . . . . . . . . . . . . . . . . . . . . . . . . . . . . . . . . . . . . . . . . . . . . . . . . . 8

         P4 enp distal segment with 3 setae . . . . . . . . . . . . . . . . . . . . . . . . . . . . . . . . . . . . . . . . . . . . . . . . 20

8.    P1 enp-2 with 4 setae . . . . . . . . . . . . . . . . . . . . . . . . . . . . . . . . . . . . . . . . . . . . . . . . . . . . . . . . . . . . . . 9

         P1 enp-2 with 3 setae . . . . . . . . . . . . . . . . . . . . . . . . . . . . . . . . . . . . . . . . . . . . . . . . . . . . . . . . . . . . . 14

9.    Caudal rami length/width ratio > 3.5 . . . . . . . . . . . . . . . . . . . . . . . . . . . . . . . . . . . . . . . . . . . . . 10

         Caudal rami length/width ratio < 3 . . . . . . . . . . . . . . . . . . . . . . . . . . . . . . . . . . . . . . . . . . . . . . . . 11

10.    Caudal rami length/width ratio > 4; female A1 8-segmented . . . . . . . . . . . . . . . . . . . . *Z. anneae* Koller and George, 2011

         Caudal rami length/width ratio > 3.5; female A1 7-segmented . . . . . . . . . . . *Z. paratypica* Becker and Schriever, 1979

         Caudal rami length/width ratio > 3.5; female A1 6-segmented . . . . . . . . . . . . . . . . . *Z. major* Sars G.O., 1919

11.    Female A1 8-segmented . . . . . . . . . . . . . . . . . . . . . . . . . . . . . . . . . . . . . *Z. carsteni* Pointner, 2017

         Female A1 6-segmented . . . . . . . . . . . . . . . . . . . . . . . . . . . . . . . . . . . . . . . . . . . . . . . . . . . . . . . . . 12

12.    Caudal rami length/width ratio about 3; female P5 with 8 setae . . . . . . . . . . . . . . . . . . . . 13

         Caudal rami length/width ratio about 2; female P5 with 7 setae . . . . . . . . *Z. pacifica* Fiers, 1991

13.    Male P5 with 7 setae . . . . . . . . . . . . . . . . . . . . . . . . . . . . . . . . . . . . . . . . . . . *Z. typica* Boeck, 1873

         Male P5 with 6 setae . . . . . . . . . . . . . . . . . . . . . . . . . . . . . . . . . . . . . . . . . . *Z. gisleni* Lang, 1948

14.    Caudal rami length/width ratio > 4 . . . . . . . . . . . . . . . . . . . . . . . . . . . . . . . . . . . . . . . . . . . . . . 15

         Caudal rami length/width ratio < 4 . . . . . . . . . . . . . . . . . . . . . . . . . . . . . . . . . . . . . . . . . . . . . . 16

15.    Female A1 8-segmented . . . . . . . . . . . . . . . . . . . . . . . . . . . . . . . . . . . . . . *Z. eliasi* Pointner, 2017

         Female A1 6-segmented . . . . . . . . . . . . . . . . . . . . . . . . . . . . . . . . . . . . . . . . *Z. erythraea* Por, 1967

16.    Female A1 7-segmented . . . . . . . . . . . . . . . . . . . . . . . . . . . . . . . . . . . . . . . . . . . . . . . . . . . . . . . . 17

         Female A1 6-segmented . . . . . . . . . . . . . . . . . . . . . . . . . . . . . . . . . . . . . . . . . . . . . . . . . . . . . . . . 18

17.    Female P5 with 9 setae . . . . . . . . . . . . . . . . . . . . . . . . . . . . . . . . . . . . . *Z. atlantica* Bodin, 1968

         Female P5 with 8 setae . . . . . . . . . . . . . . . . . . . . . . . . . . . . . . . . . . . . . . . . *Z. thistlei* sp. nov.

         Female P5 with 7 setae . . . . . . . . . . . . . . . . . . . . . . . . . . . . . . . . . . . . . *Z. valida* Sars G.O., 1919

18.    Caudal rami length/width ratio about 3 . . . . . . . . . . . . . . . . . . . . . . . . . . . . . . . . . . . . . . . . . 19

         Caudal rami length/width ratio about 2 . . **.** . . . . . . . . . **.** . . . . . . *Z. mediterranea* Lang, 1948

19.    Female P5 with 9 setae . . . . . . . . . . . . . . . . . . . . . . . . . . . . . . . . . . . . . *Z. paramajor* Bodin, 1968

         Female P5 with 8 setae . . . . . . . . . . . . . . . . . . . . . . . . . . . . . . . . . . *Z. bergensis* Drzycimski, 1968

20.    P1 enp-2 with 4 setae . . . . . . . . . . . . . . . . . . . . . . . . . . . . . . . . . . . . . . . . . *Z. montagnai* sp. nov.

         P1 enp-2 with 3 setae . . . . . . . . . . . . . . . . . . . . . . . . . . . . . . . . . . . . .**.**. . . . . . . . . . . . . . . . . . . . 21

21.    Female P5 with 7 setae; A2 exp-3 with 4 setae . . . . . . . . . . . . . . . . . . . . . . . . . . *Z. destituta* Kim J.G., Jung and Yoon, 2016

         Female P5 with 10 setae; A2 exp-3 with 3 setae . . . . . . . . . . . . . . . . . . . . *Z. tunnelli* sp. nov.

## 4. Discussion

### *4.1. World Distribution of Zosimeidae*

    Koller and George [7] summarized the information on the distribution of 15 species of *Zosime*. They argued that the small number of known species in this family is probably due to the lack of collected samples, rather than the rarity of this taxon itself. This hypothesis was probably made because *Zosime* was found more often in the deep sea than other taxa.

    In this study, the distribution for all species of Zosimeidae, including the new species, is summarized. There have been no reports of the family Zosimeidae in the South Pacific,

Indian, Arctic, and Antarctic seas. The sediment types of habitats varied, including mud, sand, biogenic carbonate sediment, and algae.

In addition, in terms of water depth, rather than preferring habitats in a specific water layer, the distribution range was found to be very wide, ranging from littoral algae rinsing samples or shallow waters (10 m) to deep sea samples (> 5800 m depth). As can be seen from Table 5, the ratio of the species distributed in the deep sea (>200 m depth) and the shallow sea are about half and half. There are 17 species found in the deep sea, of which 11 have been found in deep sea below 1000 m. There were four species showing the distribution depth across the shallow and deep seas. There are ten species distributed only in shallow waters.

**Table 5.** Zosimeid species list. *Z. rubra* Thompson I.C., 1888 (nomen nudum), is not included.

| Genus | Species | Locality | Body Length (μm) (*: mean length) | Depth (m) |
|-------|---------|----------|-----------------------------------|-----------|
| *Acritozosime* Kim, JG and J Lee, 2021 | *A. spinesco* Kim, JG and J Lee, 2021 | Northwestern Pacific, East Mariana Basin; Philippine Basin of the Philippine Sea | 406–535 (* 479) | 5078–5856 |
| *Heterozosime* Kim, JG and J Lee, 2021 | *H. tenuis* Kim, JG and J Lee, 2021 | the Southern Sea of Korea | 473–631 (* 569) | 78–117 |
| *Pseudozosime* Scott T., 1912 | *P. browni* Scott T., 1912 | North-east Atlantic Ocean, South Orkney Islands | 950 | - |
| *Peresime* Dinet, 1974 | *P. abyssalis* Dinet, 1974 | South-east Atlantic Ocean, Cape Basin. Vase blanche à globigérines | 500 | 4100 |
| | *P. pryorae* sp. nov. | North-west Atlantic Ocean, Gulf of Mexico | 376 | 1010 |
| | *P. reducta* (Becker and Schriever, 1979) | North-east Atlantic Ocean, Iberian deep sea | 375 | - |
| *Zosime* Boeck, 1873 | *Z. anneae* Koller and George, 2011 | North-east Atlantic Ocean, Great Meteor Seamount | 535–600 | 292–4015 |
| | *Z. atlantica* Bodin, 1968 | North-east Atlantic Ocean, Gulf of Gascogne (France) | 770 | 1200 |
| | *Z. bathyalis* Por, 1967 | Red sea, Gulf of Elat (Israel) | 380–400 | 180–700 |
| | *Z. bergensis* Drzycimski, 1968 | North Atlantic Ocean, Korsfjord, Raunefjord (Norway); Porcupine Seabight (Atlantic Ocean) | 450–560 | 155–512 |
| | *Z. carsteni* Pointner, 2017 | North-east Atlantic Ocean | 551.6–697.6 (* 595.4) | 284–339 |
| | *Z. changi* Kim J.G. and Lee J., 2019 | the Southern Sea of Korea | 552–680 (* 614) | 61 |
| | *Z. comata* Kim J.G. and Lee J., 2019 | the Southern Sea of Korea | 674–839 (* 769) | 96 |
| | *Z. destituta* Kim J.G., Jung and Yoon, 2016 | the Southern Sea of Korea, Off Hansando Island | 667 | 10–15 |
| | *Z. eliasi* Pointner, 2017 | North-east Atlantic Ocean | 667.0–767.1 (* 718.1) | 284–339 |
| | *Z. erythraea* Por, 1967 | Red sea, Gulf of Elat (Israel) | 420–440 | 180–190 |
| | *Z. gisleni* Lang, 1948 | North Atlantic Ocean, Gullmarfjord (Sweden); Red sea, Gulf of Elat (Israel); Baltic Sea, Kattegat (Germany) | 480–600 | 20–300 |

**Table 5.** *Cont.*

| Genus | Species | Locality | Body Length (μm) (*: mean length) | Depth (m) |
|---|---|---|---|---|
| | *Z. gymnokosmosa* Kim J.G. and Lee J., 2019 | the Southern Sea of Korea | 352–451 (* 409) | 109 |
| | *Z. major* Sars G.O., 1919 | North Atlantic Ocean, Korshaven, Risør (Norway); Gullmarfjord (Sweden) | 600–700 | 20–92 |
| | *Z. mediterranea* Lang, 1948 | Mediterranean Sea, Castiglione (Algeria) | - | Littoral |
| | *Z. pacifica* Fiers, 1991 | North-east Pacific Ocean, Santa Maria Basin, California (USA); North Atlantic Ocean, Porcupine Seabight | 420 | 50–565 |
| | *Z. paramajor* Bodin, 1968 | North-east Atlantic Ocean, Gulf of Gascogne (France); North-west Atlantic Ocean, off North Carolina (USA), North Atlantic Ocean, Porcupine Seabight | 445 | 900–3000 |
| | *Z. paratypica* Becker and Schriever, 1979 | North-east Atlantic Ocean, Iberian deep sea | 610 | 3920 |
| | *Z. reyssi* Dinet, 1974 | South-east Atlantic Ocean, Cape Basin | 800 | 3694 |
| | *Z. typica* Boeck, 1873 | North Atlantic Ocean, Oslo Fjord, Farsund, Risør (Norway), Gullmarfjord (Sweden) | 550 | 29–70 |
| | *Z. valida* Sars G.O., 1919 | North Sea/Atlantic Ocean (UK, Norway, Sweden); Baltic Sea (Germany, Sweden); Eastern Mediterranean Sea (Israel), White Sea (Russia) | 630–700 | 20–100 |
| | *Z. incrassata bathybia* Bodin, 1968 *Z. incrassata incrassata* Sars G.O., 1910 | Lyngdal Fjord, Bergen (Norway, Atlantic Ocean); Gullmarfjord (Sweden, Atlantic Ocean); off North Carolina (USA, Atlantic Ocean); Gulf of Gascogne (France, Atlantic Ocean); Kvarneric (Croatia, Adriatic Sea) | 550 | 40–3940 |
| | *Z. montagnai* sp. nov. | North-west Atlantic Ocean, Gulf of Mexico | 535 | 1010 |
| | *Z. tunnelli* sp. nov. | North-west Atlantic Ocean, Gulf of Mexico | 452 | 179 |
| | *Z. thistlei* sp. nov. | North-west Atlantic Ocean, Gulf of Mexico | 328 | 1240 |

As Seifried [16] also mentioned "As *Zosime mediterranea* Lang, 1948 was found between algae, Zosimidae is not a deep-sea taxon.". Although there are many cases of Zosimeidae found in the deep sea [7,17–20], it is difficult to distinguish it as a deep-sea taxon.

We tried to investigate the relationship between the body length of the species included in family Zosimeidae and the depth of the habitat of the species, but there was no significant correlation seen for the entire species of this family. However, in the case of Zosimeid species with a small body length of less than 440 μm, it was found that the smaller the body, the deeper it was found. In addition, most of the cases showing a relatively longer body length of 630–770 μm or more were found at a depth deeper than 400 m. Overall, the longer species were found relatively few times in deep sea depths greater than 1000 m.

Lee et al. [21] have argued that many harpacticoids found in the Gulf of Mexico tend to have a smaller size than their congeners from other regions. It has also been reported that the size of the species inhabiting this area is rather small because of the low availability of food in the area [2]. A recent study [22] summarized the composition and type of sediment by depth in the southern Gulf of Mexico area, and it was confirmed that the rate of sand was lower, and the rate of clay was higher in the deep sea than in the shallow sea. Since the grain size of the deep-sea sediments is small and the habitat will be narrowed, it can be inferred that the size of the copepods living there will be relatively small.

As mentioned in the introduction, according to the results of ecological studies conducted in this area, it is reported that Tisibidae is the dominant family [3]. Because many species of Zosimeidae previously belonged to Tisibidae, although small in body size, this taxon may be an important taxon in the Gulf of Mexico benthic ecosystem.

*4.2. Status of Gene Data and Molecular Phylogenetic Analysis with Related Taxa*

The gene sequence studies conducted so far in the family Zosimeidae are summarized in the Table S3. The Zosimeidae sequences uploaded to the NCBI database are from three studies resulting in 47 sequences, the gene sequence information corresponds to a total of five markers, of five to seven species of two genera (*Zosime, Pseudozosime*). Only six of the uploaded Zosimeidae sequences have been identified to the species level, all are mtCOI sequences of *Zosime atlantica*. The marker with the most data is 18S rRNA, and all sequences are relatively short fragment sequence information of 740 bp, except one.

In this study, an attempt was made to secure the gene sequences of the new species, but it was difficult because they were from formalin-fixed samples. Although the phylogenetic analysis using the data uploaded to NCBI is very limited, we tried to confirm the molecular phylogenetic relationship based on the commonly uploaded 18S rRNA fragment sequences for each family. In addition, based on the recorded taxonomic shifts in the Zosimeidae species, the families determined to be morphologically related (Pseudotachidiidae, Idyanthidae, Tisibidae, Ectinosomatidae, and Aegisthidae) were intensively included. To increase the accuracy of the results, only data longer than 1000 bp among uploaded 18S rRNA sequences were used.

As a result (Figure 13), it was confirmed that Ectinosomatidae, Idyanthidae, and Tisibidae showed a relatively close relationship with Zosimeidae in the order. In Easton and Thistle [23], a phylogenetic tree was created based on short fragment sequences (740 bp), and the results also confirmed a close phylogenetic relation between Zosimeidae, Ectinosomatidae, and Idyanthidae. Seifried [16] defined Idyanthidae and Zosimeidae together as Idyanthidimorpha, having a close morphological relationship with Ectinosomatidae, and claimed that three subfamilies of the previously known family Tisbidae show a distant phylogenetic relationship.

The phylogenetic tree of this study included three species belonging to subfamily Tisbinae Stebbing, 1910, and one species each from Idyanthidae and Zosimeidae. Unlike the results of Seifried [16], the results of this study showed that they were closely related. However, since the number of species included in this study was very small, more gene sequence studies from more species are needed to make more accurate claims and judgments, which should be able to fill the gap in the phylogenetic tree and increase resolution.

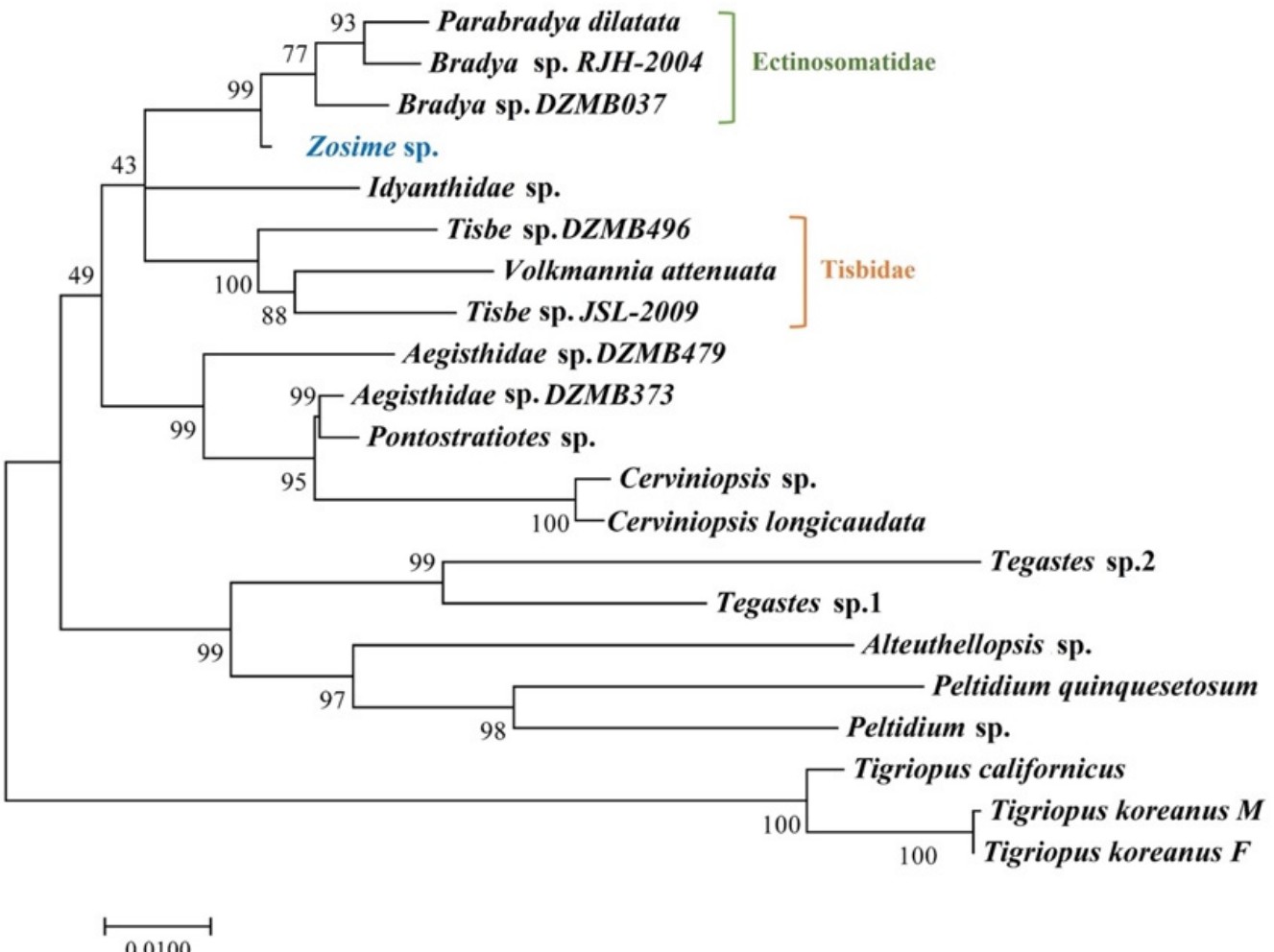

**Figure 13.** Maximum likelihood phylogenetic tree including Zosimeidae and other families based on 18S rRNA sequences.

## 5. Conclusions

Based on the morphological characters we report three new species of *Zosime* and a new species of *Peresime* from the Gulf of Mexico. The present study is the first report of Zosimeidae from the study area and suggests high zosimeid biodiversity in the Gulf of Mexico.

A key for the genera of the family Zosimeidae and keys to the species to *Zosime* and *Peresime* were provided, and the distribution and molecular research status of each species were summarized.

To infer the molecular phylogenetic relationship between Zosimeidae and the morphologically and taxonomically related families, NCBI uploaded data were used, and as a result, it was confirmed that they were closely related to Ectinosomatidae, Idyanthidae, and Tisbidae. For accurate molecular phylogenetic studies, future molecular studies with more molecular markers and various species will be needed.

As such, this study comprehensively summarizes the morphology, DNA studies, and distribution of Zosimeidae, and these contents will be used as basic data for future Zosimeidae studies or deep-sea copepods research in the Gulf of Mexico.

**Supplementary Materials:** The following supporting information can be downloaded at: https://www.mdpi.com/article/10.3390/d14030198/s1, Table S1: List of valid species of *Zosime* Boeck, 1873 with some of the morphological features based on females. Table S2: GenBank numbers of

sequences used in phylogenetic analysis in this study. Table S3: NCBI uploaded sequences list of Zosimeidae and their length (bp). References [24–28] are cited in the supplementary materials.

**Author Contributions:** Conceptualization, W.L. and M.R.L.; formal analysis, W.L. and J.Y.; investigation, W.L. and J.Y.; resources, M.R.L.; writing—original draft preparation, W.L. and J.Y.; writing—review and editing, W.L., M.R.L. and J.Y.; visualization, W.L. and J.Y.; project administration, W.L. and M.R.L.; funding acquisition, W.L. All authors have read and agreed to the published version of the manuscript.

**Funding:** This research was funded by Basic Science Research Program through the National Research Foundation of Korea (NRF) funded by the Ministry of Education, grant number 2021R1I1A2043807, by National Marine Biodiversity Institute of Korea (MABIK), grant number 2022M01100, and by the Korea Polar Research Institute (KOPRI), grant number PE22900.

**Institutional Review Board Statement:** Not applicable.

**Informed Consent Statement:** Not applicable.

**Data Availability Statement:** The voucher specimens of the species examined in the present study were deposited to National Marine Biodiversity Institute of Korea (MABIK).

**Acknowledgments:** The authors would like to express gratitude to anonymous reviewers, the Biodiversity lab (Hanyang University, Korea) members, and Paul A. Montagna (Harte Research Institute for Gulf of Mexico Studies, Texas A&M University—Corpus Christi). The sampling was a part of the Center for the Integrated Modeling and Analysis of Gulf Ecosystems (C-IMAGE) II consortium research program.

**Conflicts of Interest:** The authors declare no conflict of interest. The funders had no role in the design of the study; in the collection, analyses, or interpretation of data; in the writing of the manuscript, or in the decision to publish the results.

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
