# Peer review of "Four New Species of Zosimeidae (Copepoda: Harpacticoida) from the Southwestern Gulf of Mexico"

_diversity, doi:10.3390/d14030198_

Round 1

Reviewer 1 Report

The manuscript "Four new species of Zosimeidae (Copepoda: Harpacticoida) from the Southwestern Gulf of Mexico" by Jisu Yeom and colleagues describes new species of Harpacticoida found in the southwestern Gulf of Mexico, followed by their thorough morpholoical and molecular phylogenetic analysis and provides a a key to species of Zosimeidae.

The topic of the manuscript is interesting and worth to be considered for publication. It provides valuable results and seems to be very useful for readers.

The manuscript is written in a clear, well-structured manner; the provided morphological descriptions are extensive and detailed, with great illustrations. The morphological analysis is supported by molecular data and the revision of the whole family of the investigated species.

I strongly recommend this paper for the publication.

Author Response

Dear Reviewer,

Thank yo so much for your encouragement. We have updated the manuscript based on the mutual comments, and new version of illustrations with better resolution.

with best regards.

Wonchoel Lee

Reviewer 2 Report

Recent studies on the dep-sea harpacticoid fauna from the Americas have shown that, despite the large number of harpacticoid species that have been reported from the Gulf of Mexico in grey literature, only two species have been formally described from that region. These two species have been described from the US Gulf of Mexico, and none had been described from the Mexican Gulf of Mexico. This manuscript is a nice piece of research and it is valuable since four new species of the family Zosimeidae are described from the Mexican Gulf of Mexico, being these the first published and documented records of harpacticoid species whose type localities are within the boundaries of the southern Gulf of Mexico. These finds will be very useful for environmental monitoring in this part of the Gulf of Mexico. Additionally, the authors added some extra value to their manuscript with a key to the genera of the family, and with a key to the species of the genera Peresime and Zosime. The manuscript is well written and the figures are of excellent quality. However, I suggest preparing plates with less but larger figures, especially for the mouth parts and other parts as indicated in the text. I am not a native English speaker, but took the liberty to suggest some corrections to improve the readability of the text. Briefly, I suggest adopting a more telegraphic style and deleting uneccessary words, The text is too long because it contains the full description of four species, but the reader will benefit from a more concise and clear text. There are some points that, in my opinion, the authrs should address. Recently, a book on deep-sea crustaceans of the Americas was published by Springer. There's a chapter on deep-sea harpacticoids from the Americas with useful information on the deep-sea harpacticoid fauna from the Gulf of Mexico. It would be a good idea to acknowledge the project (and the project's head) that provided the samples. Some comments were done to the Introduction and Materials and Methods in the main document. As for the taxonomic part, I added some additional comments. Briefly, I suggest adding a line with the type species, and another line with the other specis of each genus just after the genus name Zosime and Peresime as indicated in the text. The genus name Zosime is feminine, and I suggest checking the species names of the new species of Zosime. The gender of the species names should agree in gender with that of the genus. Note also that Peresime pryori was dedicated to a woman, and therefore, the ending "i" in the species name is incorrect. The correct species name should be "pryorae". Also, it is important to add the type locality for each species, including the coordinates, collector's name, date of colection, etc., and any other environmental information available. I suggest a DIFFERENTIAL DIAGNOSIS instead of a DIAGNOSIS for each species. These differential diagnoses should contain only those characters useful for species separation, i.e., those characters not present in any other species of the genus. I suggest checking the use of some terms such as "anterior" and "abexopodal"; some comments on this appears in the text. The use of hyphens and N-dashes should also be checked. The REMARKS for each species need to be improved also. It seems to me as if the authors based their conclusions on the relationships of each species based only on some identification keys. However, identification keys do not reflect the phylogeny, and should not be used to suggest any resemblance between the species. For this, the authors shoud prepare a table with the most important characters and character states to unravel the relationships between their new species and their congeners. Such table would be of great value for future studies. Additional comments and corrections appear in the main document.

Author Response

Dear Reviewer,

We thank you very much for your valuable comments and remarks which have helped us to upgrade the manuscript.

We believe that the improved version of the manuscript is now suitable for publication in Diversity.

All but a few things have been reflected in terms of correcting words, changing the arrangement, or correcting typos in MS. Thank you for your meticulous correction.

Below is the answer to remark in pdf file.

<Abstract>

“Z. destituta Kim J.G., Jung & Yoon, 2016” - I'm not sure if these initials are necessary.

 We marked it as shown in WoRMS.

setal formula, female P5

 We modified to “setal armature formula and shape of female P5 and caudal rami”.

“The type of P2 setae”- Please be more specific. Does this refer to the ornamentation of the setae?

 We modified to “the length of P2 setae”.

<Introduction>

“The Gulf of Mexico has undergone two major oil spills.” - This is ok. Some infornation relevant to the present manuscript can be found in Gómez, S. & Rivera-Sánchez, KI (2020) The deep-water benthic Harpacticoida (Copepoda) of the Americas, p. 47-125. In: Hendrickx, ME (ed.) 2020. Deep-sea pycnogonids and curstaceans of the Americas. Springer.

 Thank you for recommending good literature.

“has been studied more, with regards ecological monitoring, than other seas. Several studies have been conducted on planktonic and benthic organisms in the Gulf of Mexico [1–5].

”- Please add a reference where the reader can find some more infornation on this.

 We marked the reference.

Please rephrase. Try with more simple sentences. I suggest: "The Gulf of Mexico has undergone two major oil spills and has been the subject of several environmental studies focused on planktonic and benthic organisms [1-5]."

 We changed it by reflecting the revision made by the reviewer.

“The number of harpacticoid copepods reported in the Gulf of Mexico is small, with only 71 species listed [3]. However, a total of 696 harpacticoid species were identified 30 during a survey in the Northern Gulf of Mexico [2] implying they have a high diversity in the deep-sea” - Add a reference for this.

 We marked the reference.

“a classification key encompassing all species of the family, - ”a key to the genera of the famiy and to the species of each genus”

 We changed it by reflecting the revision made by the reviewer.

<Materials and Methods>

Please give the name of the project and the head's name in the Acknowledgements.

 We added the information in the Acknowledgements.

Guess this is one of the two major oil spills referred to in the Introduction. It would be a good idea to add some more information on this oil spill, and the respective references.

 We added some information on this oil spill in sampling methods part, and also another oil spill information in introduction.

“by washing the samples over 45 μm ” - This is not clear. The sediment saples were sieved using a 45 µm mesh sieve? Please be more specific and clear.

 Thank you for the remark. We modified.

“The descriptive terminology of Huys et al. [8] was adopted.” - This is ok. Just curious. Why not Huys & Boxshall (1991) (Coppeod evolution) and Seifried (2003) (Phylogeny of Harpacticoida (Copepoda): revision of "Maxillipedasphalea" and Exanechentera)?

 It is a reference to abbreviations that has been commonly used in many harpacticoid papers.

2.3. Phylogenetic analysis - I suggest adding the aims of these analyses somewhere. Also, is the matrix available?

 The last part of the discussion (1.2.) contains information related to this analysis.

Does the matrix you're asking mean the fasta file?

Because we wrote down the analysis method and provided the accession numbers of the sequences used as a supply table, we decided that it would be enough. However, we will provide the fasta file if it is essential.

<Results> 

“Zosime” - I'm not sure where the genus name Zosime comes from, but the Latin suffix "ime" is feminine and the gender of the specific epithet of the type species (Z. typica) is feminine. Therefore, the genus name Zosime should be treated as feminine (ICZN art. 30.1.1). I'm not sure if the masculine ending "i" for the three species described here is correct in terms of ICZN art. 31.1 (agreement in gender with the generic name). Please check this and make the appropriate amendments.

 Thank you for the important revision.

Even though the gender of the genus is not known for sure, among the names of congeners in Zosime, there are already species names that end with '-i', which are named after humans [e.g. Zosime changi Kim J.G. & Lee J., 2019 / Zosime carsteni Pointner, 2017 / Zosime eliasi Pointner, 2017]. Therefore, according to ICZN art. 31.1.2, we named by adding ‘-I’ to the person's name. Since only the species name of Peresime is derived from the female name, we will revise it to '-ae'.

(31.1.2. A species-group name, if a noun in the genitive case (see Article 11.9.1.3) formed directly from a modern personal name, is to be formed by adding to the stem of that name -i if the personal name is that of a man, -orum if of men or of man (men) and woman (women) together, -ae if of a woman, and -arum if of women; the stem of such a name is determined by the action of the original author when forming the genitive.)

I suggest adding a separate paragraph with a list of all the valid species of Zosime, and another line with the type species.

 As reviewer suggested, we have inserted information on the type species.

However, since all other species are shown in classification key and the table that summarizes the distribution information, it is overlapping to provide a separate list.

Material Examined – Is this the sampling station? If so, this should be moved to the Type Locality. Add a separate paragraph with the Type Locality.

 Thank you. A separate paragraph related to Type Locality were inserted, and information on the collection site of holotype was entered.

“The specific name refers to” - I suggest: "The new species is dedicated to....for his excellent....." Also, please add: "it is a noun in the genitive case, gender masculine." if appropriate, because the gender of the genus name Zosime is feminine (see my comments to the genus name Zosime above).

 As you advised, we revised the sentence and added it.

Diagnosis – I suggest preparing a shorter differential diagnosis with those characters that separates the new species from all other species.

 As you suggested, I have kept only the characteristic contents of the species and summarized them.

Please reword. I suggest "..., covering anal somite in both sexes."

 We reflected the revision made by the reviewer in MS.

These two bars are of different width, but of the same length. I suggest uniformizing the width of the scale bars. Also, one of these bars can be deleted since they are of the same length.

 We modified the scale bar.

“Posterior margins of each urosomite serrated with rough denticles.”

Does this refer to all urosomites or to he genital double somite? If this is for all urosomites, the same appears in the first line. Please check and correct.

 It was deleted because it was duplicated.

A larger drawing of the CR would be nice.

 We adjusted the size of Figure 2.

"anterior" - Proximal? Inner? the term "anterior" is confusing here.

 We changed to Proximal.

“with one scar equivalent to an abexopodal seta missed during dissection” - The seta(e) on the endopodal segment(s) are not abexopodal setae. Abexopodal setae are found only on the basis or alobasis and are oposite to the exopod. Also, the scar is not "equivalent" to the seta. The term "equivalent" is misleading. The scar indicates the position of a seta that was lost during dissection. Also, there seems to be two scars, one proximal and one subdistal.

 We changed to “with a subdistal seta missed during dissection and sample preparation (The scar indicates the position of a seta.)”

“row of spinules around outer lateral margin;” –

I cannot see these spinules. What I see is a proximal inner spinule.

 Deleted

“two geniculate spines”

These setae are broken in figure and the geniculations cannot be seen.

 Yes. Although not visible in this figure, the geniculations are identified prior to specimen damage.

“two stout pinnate and naked spine, ”

Pinnate and naked? THis is contradictory. Please check. Two stout pinnate setae, and one naked spine?

 We changed to “two shout pinnate spines and naked spine.”

For the description of the situation of the setae on the second endopodal segment I suggest the terms: inner lateral, inner distal, distal, and outer distal.

 Thanks for the good advice. Some corrections have been made.

“pinnate seta” - Epipodite? (fig. 4b)

 We modified.

“produced into strong curved claw” – The claw seems to be articulated with basis...this claw should be fused to allobasis. Please check. (fig. 4c)

 We checked and modified the figure.

Check on the use of hyphens VS N-dashes, and correct the entire document if needed.////

legs 1–4: Ok....this is an N-dash that is normally used for intervals (pages, figures, etc,). It depends on the journal's format, but if needed, please check the entire document and replace hyphens with N-dashes where appropriate.

 Thanks for the important point. Overall edited.

“enp-3 reached only distal 1/3 of exp-3” - Move this to the description of the endopod.

“endopod only reached to proximal 1/3 of exp-3” - The same as in P2.

“endopod reached to proximal region 201 of exp-3” - The same as in P2 and P3.

 We moved as you advised.

Please check. There are two inner setae in the written description.

 We changed to 1.220

“slightly depressed and rectangular rather than triangular.” - with parallel lateral margins?

 We have corrected the description.

“superficially” - Not sure what do the authors mean with this.

 We changed to “morphologically”

“has less setae” - Please explain and be more specific.

 Based on your advice, we have modified it in a way that's clearly marked.

“a more progressively reduced form.” –

comparatively more derived than Z. paratypica? Is this what the authors mean?

 As you understand it. We changed the sentence.

“smooth” - rounded?

 We have reflected your corrections.

proximal? inner? The term "anterior" is confusing.

 We changed to “inner”.

“and several spinules along outer distal margin.” - These are not visible in figure 8D.

 Deleted.

“P6 damaged” - Not shown?

 We could not provide figure.

“(G) P5 and P6.” - Add "H" for P6 and make the necessary corrections to the text.

 It has been corrected.

“and genital field.” - And segmentation of urosome? Check the other descriptions.

 We changed to “segmentation of urosome”.

proximal? inner? The term "anterior" is confusing.

 We changed to “inner”.

The description of this species follows the same format as in the previous species, and have therefore, the same flaws. Please correct this description based on the corrections made to the previous ones. Try to adopt a more telegraphic style and try to be more clear and specific. This is a very good piece of research, but it is too long and the reader will undoubtedly benefit from a more concise document.

 Based on reviewer’s meticulous corrections, the description of this species has also been revised. Thanks.

“Scale bars: (A) 100 μm; (B-D) 10 μm.” - But B and B' have different scale bars. Please correct.

 We changed to “B, B’-D”

“with rough denticles” - deeply serrate?

 It has been corrected.

“anterior “- Proximal? The term "anterior" here and in te other descriptions is miselading. The spinules on the first segment of A1 are situated ventrally on the inner medial margin of the segment.

 We changed to “proximal”.

“abexopodal seta “- The term "abexopodal" refers to the seta on the basis situated oposite to the exopod.

 We changed to “seta”

Fig.11 A1 - Add a caption for segments indicated with II, III, V, VI and VII

 We added the caption in Fig. 11.

Please be consistent throughout the text: three- or 3-? Check the text for similar errors.

 To make it easier to use the identification key, it is marked with Arabic numbers only within the key. If it all must be unified, we will fix it.

Please describe the position of the spinules more accurately. Check the other descriptions for similar errors.

 We improved the description.

At the base of the outer seta?

 Modified. 

“small denticle-like shape” - This is not clear.

 We changed to “small denticle shaped spine”.

<Discussion>

[Line 724: lack of collected samples  low sampling effort]

 The amount of samples is often not proportional to the amount of sampling effort. Therefore, we would like to leave it as what we originally wrote.

“Habitat sediment types were varied,” - Please reword.

 We changed to “The sediment types of habitats varied”.

“in terms of ratio, the species distributed in the deep sea (deeper than 200 m) and the shallow sea are about half and half. ” - Please rephrase.

 We changed to “the ratio of the species ~ “.

Line 743 “the species”- Of the five genera?

 Yes. we modified the sentence.

Line 746~747 - But above the authors said that they did not find any correlation.....// Please reword.

 We modified the sentence.

Line 756 “it can be inferred that the size of the copepods inhabiting there would be smaller.” - Why? Any reference or analysis to support this?

 This sentence has been modified because it is the opinion of the author.

Studies have shown that smaller grain size of sediments generally form narrower habitats, and thus smaller nematodes outnumber harpacticoids (Coull 1985).

Line 758 “The four species newly reported in this study are also smaller than other species in the family,” -But, for example, Z. montagnai was descvribed after 2 females and 1 male and the length of the holotype was the only measuremente reported here. Z. thistlei was described from 1 female and 1 male, and Z. tunnelli was described after 1 female only.  Peresime poyri was described only after one female....what about the intraspecific variabiity in body length? This can be assessed only by measuring as many specimens of each sex as possible.

 Thanks for your comments. As you said, the number of individuals that were able to measure their length was very small. The paragraph was deleted because it was judged that there is a possibility of intraspecific variation.

“Easton,E.E. & Thistle,D.” - Please check how the references must be cited.

 We changed to “Easton and Thistle”.

<Additional comments in Word file>

I suggest preparing plates with less but larger figures, especially for the mouth parts and other parts as indicated in the text.

 We have resized the figures to be larger.

I suggest adopting a more telegraphic style and deleting uneccessary words, The text is too long because it contains the full description of four species, but the reader will benefit from a more concise and clear text.

--> We were able to change the MS to make it better by removing unnecessary expressions and reflecting your modifications. thank you.

The REMARKS for each species need to be improved also. It seems to me as if the authors based their conclusions on the relationships of each species based only on some identification keys. However, identification keys do not reflect the phylogeny, and should not be used to suggest any resemblance between the species.

For this, the authors shoud prepare a table with the most important characters and character states to unravel the relationships between their new species and their congeners. Such table would be of great value for future studies.

----> Thanks for the good revision. The remarks of four species were revised and improved. In ‘Remarks’, it was not intended to include phylogenetic content based on the identification key, but to consider situations that researchers can actually encounter when they follow the identification key and include comparisons with morphologically similar species.

Additionally, a table suggested by reviewer was prepared and added as supplementary table. We agree that it will be helpful in future related research. In fact, preparing this table also helped improve diagnosis.
